# Diffusion-Based Probabilistic Uncertainty Estimation for Active Domain Adaptation

**Zhekai Du,  Jingjing Li**[*]
School of Computer Science and Engineering
University of Electronic Science and Technology of China
zhekaid@std.uestc.edu.cn, lijin117@yeah.net

## Abstract

Active Domain Adaptation (ADA) has emerged as an attractive technique for assisting domain adaptation by actively annotating a small subset of target samples. Most ADA methods focus on measuring the target representativeness beyond traditional active learning criteria to handle the domain shift problem, while leaving the uncertainty estimation to be performed by an uncalibrated deterministic model. In this work, we introduce a probabilistic framework that captures both data-level and prediction-level uncertainties beyond a point estimate. Specifically, we use variational inference to approximate the joint posterior distribution of latent representation and model prediction. The variational objective of labeled data can be formulated by a variational autoencoder and a latent diffusion classifier, and the objective of unlabeled data can be implemented in a knowledge distillation framework. We utilize adversarial learning to ensure an invariant latent space. The resulting diffusion classifier enables efficient sampling of all possible predictions for each individual to recover the predictive distribution. We then leverage a t-test-based criterion upon the sampling and select informative unlabeled target samples based on the p-value, which encodes both prediction variability and cross-category ambiguity. Experiments on both ADA and Source-Free ADA settings show that our method provides more calibrated predictions than previous ADA methods and achieves favorable performance on three domain adaptation datasets.

## 1 Introduction

Machine learning algorithms heavily depend on large amounts of labeled data. However, in many real-world scenarios, the distribution of data can change over time or across different domains, which challenges the training of models to perform consistently in all situations. Unsupervised Domain Adaptation (UDA) [1, 2, 3] addresses this problem by transferring knowledge learned from a labeled source domain to an unlabeled target domain. Despite impressive achievements, UDA struggles to bridge the performance gap with fully supervised methods due to the lack of target-supervised information. Recognizing this limitation, a promising alternative is Active Domain Adaptation (ADA) [4, 5, 6], which actively annotates a small subset of target data to greatly benefit the UDA model.

In ADA, the primary concern revolves around designing a criterion for selecting the most informative target samples that can have the maximum impact on performance once annotated. Existing ADA methods have been mainly enlightened by traditional Active Learning (AL) methods [7, 8, 9, 10] and rely on criteria that consider both predictive uncertainty and representativeness of the entire distribution. While they have shown empirical success, they usually handle the distribution shift by further measuring the representativeness of the target domain through various criteria (e.g., the output of a domain discriminator [11], free energy score [5], weighted clustering [4]). However, these

---

[*]Corresponding Author.

37th Conference on Neural Information Processing Systems (NeurIPS 2023).

methods still estimate uncertainty based on metrics like confidence [12], distinctive margin [6], and entropy [4], obtained from a deterministic model. It is worth noting that distribution shift can easily lead to overconfident predictions for deep models [13], especially deterministic ones [14, 15], which results in poorly calibrated point estimates and unreliable uncertainty estimation in ADA.

In this work, we aim to accurately estimate the posterior distribution of the target variable $y$ given the input $x$ for target domain samples (i.e., $p(y|x)$), by modeling the uncertainty in both data generation and model prediction processes. To achieve this, we utilize diffusion-based generative models [16, 17, 18] to explore the implicit predictive distribution beyond a point estimate for each target domain sample. Specifically, we harness the capabilities of diffusion models for classification [19], which involve forward and reverse diffusion chains to generate predictions from the conditional predictive distribution, without any assumptions on the parametric form of $p(y|x)$. Notably, another work [20] also tackles the miscalibration issue in ADA by considering the model prediction as a distribution on the probability simplex and introducing a Dirichlet prior over predictive distributions. However, such a constrained parametric form might not be effective if the prior distribution fails to accurately capture the predictive uncertainty [19]. Additionally, their approach only focuses on modeling the probability distribution in the model prediction space, neglecting the inherent uncertainty present in the data representation from a generative perspective.

To approximate the posterior distribution of both latent data representation and model prediction, we use variational inference by optimizing the evidence lower bound (ELBO) corresponding to the log-likelihood of all data points. Concretely, we show that for labeled data, the training objective can be formulated by Variational Autoencoder (VAE) [21] and a diffusion-based classifier [19] in the latent space. For unlabeled data, they can also be effective utilized by optimizing the ELBO, which can be implemented in a knowledge distillation framework. With the formulation, we establish a two-stage training procedure where we first learn a guided classifier for mean estimate and then train a diffusion probabilistic model guided by that for predictive distribution recovery. To ensure a shared latent embedding that can be used to solve tasks for both labeled and unlabeled data, we incorporate adversarial learning [1, 22], resulting in a Diffusion-based Adversarial Probabilistic Model (DAPM). With a collection of predictions generated by DAPM for each unlabeled target sample, we conduct t-test [23] between scores of the two most predicted classes and select the most informative samples based on the p-value, which generally takes into account the sampling scale, prediction variability and the cross-category ambiguity to estimate the prediction uncertainty for active annotation.

**Our contributions:** 1) We formulate ADA in a probabilistic framework for uncertainty estimate, which leverages the abilities of VAE and diffusion-based classification models to capture distributions of both data and prediction. 2) We conduct a two-stage training procedure and use adversarial learning to ensure an invariant latent space. A t-test-based criterion is utilized to estimate informativeness from multiple aspects. 3) We show that our method can naturally handle active learning for both UDA and Source-Free DA (SFDA). Experiments on three datasets show the effectiveness of our method.

## 2    Related Work

**Active Learning** [24, 25] aims to tap the fully supervised performance by only annotating a small subset of training data. The selection of informative samples can be made using a variety of criteria. Committee-based methods [26, 27, 28, 29] leverage diverse classifiers and evaluate the data informativeness based on their disagreement. Representativeness-based methods [30, 31] aim to select samples that are diverse or representative enough of the entire training distribution, typically through clustering [32] or core-set selection [33]. Uncertainty-based methods resort to various uncertainty heuristics such as entropy [34], confidence [35], and best-vs-second-best score [36], etc, to select the most uncertain instances. However, these AL methods will fail when there is a distribution shift between labeled and unlabeled data, limiting their applications in domain adaptation.

**Active Domain Adaptation.** A series of ADA works have been conducted to select the most informative target samples under domain shift. As an early work, AADA [37] applies a domain discriminator to evaluate the domainness of target samples and weights it with entropy-based uncertainty. TQS [11] designs a transferrable query criterion based on a classifier committee and a domain discriminator to handle domain shift. CLUE [4] performs an entropy-weighted clustering to select both uncertain and diverse samples. SDM [6] uses a distinctive margin loss for training and selects data lying in the margin. EADA [5] leverages the free-energy bias and uses an energy-based criterion to evaluate

the uncertainty and domainness of target samples. Despite the empirical success, these methods mainly focus on compensating for the domain gap by incorporating target-representativeness into the query function. However, the uncertainty criteria they used are still based on a point estimate from a deterministic model, which can be easily miscalibrated for out-of-distribution data [38]. To remedy this issue, DUC [20] places a Dirichlet prior on the class probability and evaluates the predictive uncertainty with a probabilistic view. However, they only consider the uncertainty in model outputs. Besides, the true predictive distribution may not be captured by such a restricted distribution form.

**Diffusion-based generative models** have been widely appreciated for their impressive capabilities of generating high quality and diverse samples [17, 16, 39, 18]. While most works focus on high-dimensional generative tasks such as (conditional) image generation [39, 16], super-resolution [40, 41], image inpainting [42, 43], etc., some attempts have been made to apply diffusion models to categorical data [44] and discrete data [45], indicating their potential in classification tasks. Recently, Han et al. [19] utilize diffusion models to recover predictive distributions in regression and classification tasks, verifying their effectiveness for uncertainty estimation. Different from [19] that focuses on in-distribution scenarios, we consider a more challenging and realistic scenario of capturing predictive distribution in a cross-domain fashion. In addition, we also model the underlying variation of the data in low-dimensional latent space, which can be viewed as a data-level uncertainty and more favorable for downstream tasks compared to original image space [46].

## 3 Methodology

In ADA, we have a source domain $\mathcal{S} = \{(\boldsymbol{x}_i^s, y_i^s)\}_{i=1}^{n_s}$ with $n_s$ labeled samples, and a target domain $\mathcal{T} = \{(\boldsymbol{x}_i^t)\}_{i=1}^{n_t}$ with $n_t$ unlabeled samples, where $\boldsymbol{x}_i^s(\boldsymbol{x}_i^t)$ is the input instance, and $y_i^s \in \mathcal{Y}$ is the corresponding label. It is assumed that both domains share a common label space $\mathcal{Y}$ but conform to different data distributions. Domain adaptation aims to train a model $f_\Omega : \mathcal{X} \rightarrow \mathcal{Y}$ parameterized by $\Omega$ that achieves good predictive performance on the target domain. To achieve this goal, the problem of ADA involves selecting a small subset of informative samples $\mathcal{T}_l \subset \mathcal{T}$ for annotation with a budget $\mathcal{B}$, where $\mathcal{B} \ll n_t$, such that the model performance on the target domain can be maximally improved. The selection process is typically completed in multiple rounds, where in each round, the model queries $b$ samples from $\mathcal{T}_u$ (the remaining unlabeled samples in $\mathcal{T}$) and adds them to $\mathcal{T}_l$. This process is repeated until the budget is exhausted. In this work, we examine active learning for both UDA and source-free UDA [47], where, besides using unlabeled target data, we can utilize the raw source data and a source-trained model, respectively, for the purposes of selection and adaptation.

### 3.1 Preliminary of Classification Diffusion Models for Uncertainty Estimation

In AL, a major concern in sample selection is predictive uncertainty. For a $K$-way classification problem, given an input variable $\boldsymbol{x}$, the predictive uncertainty can be expressed as the posterior distribution over the predicted variable $\boldsymbol{y}$[2] after observing the training set $\mathcal{D}$, i.e., $p(\boldsymbol{y} \mid \boldsymbol{x}, \mathcal{D})$. Most AL methods merely pay attention to accuracy and adopt a non-Bayesian approach to train the model, e.g., Maximum Likelihood Estimation (MLE) or Maximum A Posteriori (MAP). As frequentist techniques, they are not capable of inferring the variance of predictive distribution $p(\boldsymbol{y} \mid \boldsymbol{x})$, but merely provide a mean estimate $\mathbb{E}[\boldsymbol{y} \mid \boldsymbol{x}]$ under the assumption of additive noise [19]. Taking a Bayesian view, the predictive uncertainty can be modeled by assuming distributions over network parameters [48, 49], while it involves expensive computation and intractable posterior inference. Some alternative methods include modeling the output of a neural network as a probability distribution over possible outcomes [50, 5] or adding the noise term in the model outputs [51]. As explicit modeling methods, they all assume a specific form in $p(\boldsymbol{y} \mid \boldsymbol{x})$ (e.g., Gaussian or Dirichlet distribution).

In this work, we leverage the deep generative models to model the implicit predictive distribution. Specifically, we start with the classification diffusion models [19]. Re-denoted by $\boldsymbol{y}_0 \in \mathbb{R}^K$ the one-hot label, the diffusion model first applies a forward diffusion process $q(\boldsymbol{y}_{1:T} \mid \boldsymbol{y}_0, \boldsymbol{x})$ to original data by iteratively perturbing it to latent representations $\boldsymbol{y}_{1:T}$ with Gaussian noises. For each step $t$,

$$q\left(\boldsymbol{y}_t \mid \boldsymbol{y}_{t-1}, \boldsymbol{x}\right) = \mathcal{N}\left(\boldsymbol{y}_t; \sqrt{1 - \beta_t}\boldsymbol{y}_{t-1} + \left(1 - \sqrt{1 - \beta_t}\right) f_\Omega(\boldsymbol{x}), \beta_t \boldsymbol{I}\right), \tag{1}$$

where $\beta_t \in (0, 1)$ is the noise variance and fixed as hyperparameter. Here the model output $f_\Omega(\boldsymbol{x})$ (mean estimate) is injected in the forward process to act as the prior knowledge about the relationship

---

[2]We use bold font to denote a probability vector: $\boldsymbol{y} = [y_1, \ldots, y_K]$.

between $\boldsymbol{x}$ and $\boldsymbol{y}_0$, so that with a sufficiently large $T$ and proper schedule $\{\beta_t\}_{t=1}^T$, we have

$$p\left(\boldsymbol{y}_T \mid \boldsymbol{x}\right) = \mathcal{N}\left(f_\Omega(\boldsymbol{x}), \boldsymbol{I}\right). \tag{2}$$

Note that the specific form of the forward diffusion process enables an efficient sampling for arbitrary steps in a closed form:

$$q\left(\boldsymbol{y}_t \mid \boldsymbol{y}_0, \boldsymbol{x}\right) = \mathcal{N}\left(\boldsymbol{y}_t; \sqrt{\bar{\alpha}_t}\boldsymbol{y}_0 + \left(1 - \sqrt{\bar{\alpha}_t}\right) f_\Omega(\boldsymbol{x}), \left(1 - \bar{\alpha}_t\right)\boldsymbol{I}\right), \tag{3}$$

with $\alpha_t := 1 - \beta_t$ and $\bar{\alpha}_t := \prod_{s=0}^t \alpha_s$. An appreciable feature of diffusion models is that if we know the posterior distribution $q(\boldsymbol{y}_{t-1} \mid \boldsymbol{y}_t, \boldsymbol{x})$, we can sample $y_T \sim \mathcal{N}(f_\Omega(\boldsymbol{x}), \boldsymbol{I})$ and run the reverse diffusion process to gradually recover the original data from $p(\boldsymbol{y}_0 \mid \boldsymbol{x})$ under the guidance of $f_\Omega(x)$. Unfortunately, $q(\boldsymbol{y}_{t-1} \mid \boldsymbol{y}_t, \boldsymbol{x})$ is actually intractable, and a feasible alternative is to use a model $p_\theta(\boldsymbol{y}_{t-1} \mid \boldsymbol{y}_t, \boldsymbol{x})$ for approximation, which can be trained with the following ELBO:

$$\log p_\theta\left(\boldsymbol{y}_0 \mid \boldsymbol{x}\right) \geq \mathbb{E}_{q(\boldsymbol{y}_{1:T}\mid\boldsymbol{y}_0,\boldsymbol{x})}\left[\log \frac{p_\theta\left(\boldsymbol{y}_{0:T} \mid \boldsymbol{x}\right)}{q\left(\boldsymbol{y}_{1:T} \mid \boldsymbol{y}_0, \boldsymbol{x}\right)}\right] := \mathcal{L}_{ELBO}^d(\boldsymbol{x}, \boldsymbol{y}_0) := \mathcal{L}_0 + \sum_{t=2}^T \mathcal{L}_{t-1} + \mathcal{L}_T, \tag{4}$$

$$\mathcal{L}_0 := \mathbb{E}_q\left[-\log p_\theta\left(\boldsymbol{y}_0 \mid \boldsymbol{y}_1, \boldsymbol{x}\right)\right], \tag{5}$$

$$\mathcal{L}_{t-1} := \mathbb{E}_q\left[D_{\mathrm{KL}}\left(q\left(\boldsymbol{y}_{t-1} \mid \boldsymbol{y}_t, \boldsymbol{y}_0, \boldsymbol{x}\right) \| p_\theta\left(\boldsymbol{y}_{t-1} \mid \boldsymbol{y}_t, \boldsymbol{x}\right)\right)\right], \tag{6}$$

$$\mathcal{L}_T := \mathbb{E}_q\left[D_{\mathrm{KL}}\left(q\left(\boldsymbol{y}_T \mid \boldsymbol{y}_0, \boldsymbol{x}\right) \| p\left(\boldsymbol{y}_T \mid \boldsymbol{x}\right)\right)\right], \tag{7}$$

where $D_{KL}$ is the Kullback-Leibler (KL) divergence. It is worth noting that scheduling small $\beta_t$ will make $q\left(\boldsymbol{y}_{t-1} \mid \boldsymbol{y}_t, \boldsymbol{y}_0, \boldsymbol{x}\right)$ still a Gaussian [52], which can be explicitly formulated using Bayes theorem. Therefore $\mathcal{L}_{t-1}$ can be evaluated in a closed form. $\mathcal{L}_T$ contains no optimizable parameter and is assumed to be sufficiently small and thus can be ignored. By adopting the appropriate form of $p_\theta(\boldsymbol{y}_{t-1} \mid \boldsymbol{y}_t, \boldsymbol{x})$ and reparameterization like DDPM [17], the model can be trained effectively by being tasked at predicting the forward noise $\epsilon$ for sampling $y_t$, with $\boldsymbol{\epsilon}_\theta\left(\boldsymbol{x}, \boldsymbol{y}_t, f_\Omega(\boldsymbol{x}), t\right)$.

### 3.2 Diffusion-Based Adversarial Probabilistic Model for ADA

The above conditional diffusion model admits the sampling of multiple predictions from $p(\boldsymbol{y} \mid \boldsymbol{x}, \mathcal{D})$ for statistical analysis. However, it assumes that the evaluated sample $\boldsymbol{x}$ and the training set $\mathcal{D}$ comes from the same distribution, which is not the case in UDA. Aware of this, we aim to build a model that induces a common latent space $\mathcal{Z}$, where the source and target samples share the same representations. We first consider the following generative process defined with the joint probability distributions (We omit the subscript $i$ to represent arbitrary data point in the dataset):

$$p\left(\boldsymbol{y}_{0:T}^s, \boldsymbol{x}^s, \boldsymbol{z}^s\right) = p\left(\boldsymbol{z}^s\right) p\left(\boldsymbol{x}^s \mid \boldsymbol{z}^s\right) p\left(\boldsymbol{y}_{0:T}^s \mid \boldsymbol{z}^s, \boldsymbol{x}^s\right), \tag{8}$$

$$p\left(\boldsymbol{y}_{0:T}^t, \boldsymbol{x}^t, \boldsymbol{z}^t\right) = p\left(\boldsymbol{z}^t\right) p\left(\boldsymbol{x}^t \mid \boldsymbol{z}^t\right) p\left(\boldsymbol{y}_{0:T}^t \mid \boldsymbol{z}^t, \boldsymbol{x}^t\right), \tag{9}$$

where $\boldsymbol{z}^s(\boldsymbol{z}^t) \in \mathcal{Z}$ is the latent representation. To model the inference process of $\boldsymbol{z}^s(\boldsymbol{z}^t)$, we employ a shared encoder parameterized by $\rho$ for both the source and target domains, supposing a domain-shared embedding. And we use two separate decoders with parameters $\phi$ and $\psi$ for the source and target reconstructions, respectively, since the generative process requires domain-specific information encoded in the parameter. Due to intractable posterior distributions, we then aim to learn latent variables for both domains using variational inference, with the following variational objectives:

**Objective for Labeled Source Samples.** For a labeled source sample $(\boldsymbol{x}^s, \boldsymbol{y}_0^s)$, our goal is to learn both the low-dimensional embedding $\boldsymbol{z}^s$ and latent class variable $y_{1:T}^s$, which can be achieved by maximizing the ELBO of the marginal log-likelihood as follows:

$$
\begin{aligned}
\log p\left(\boldsymbol{x}^s, \boldsymbol{y}_0^s\right) &= \log \int p\left(\boldsymbol{x}^s, \boldsymbol{y}_{0:T}^s, \boldsymbol{z}^s\right) d\boldsymbol{y}_{1:T}^s d\boldsymbol{z}^s \\
&\geq \mathbb{E}_{q\left(\boldsymbol{y}_{1:T}^s, \boldsymbol{z}^s \mid \boldsymbol{x}^s, \boldsymbol{y}_0^s\right)}\left[\log \frac{p\left(\boldsymbol{y}_{0:T}^s, \boldsymbol{x}^s, \boldsymbol{z}^s\right)}{q\left(\boldsymbol{y}_{1:T}^s, \boldsymbol{z}^s \mid \boldsymbol{x}^s, \boldsymbol{y}_0^s\right)}\right] \\
&= \underbrace{\mathbb{E}_{q_\rho\left(\boldsymbol{z}^s \mid \boldsymbol{x}^s, \boldsymbol{y}_0^s\right)}[\log p_\phi(\boldsymbol{x}^s \mid \boldsymbol{z}^s)] - D_{KL}\left(q_\rho\left(\boldsymbol{z}^s \mid \boldsymbol{x}^s, \boldsymbol{y}_0^s\right) \| p(\boldsymbol{z}^s)\right)}_{\text{VAE}} + \\
&\quad \mathbb{E}_{\boldsymbol{z}^s \sim q_\rho\left(\boldsymbol{z}^s \mid \boldsymbol{x}^s, \boldsymbol{y}_0^s\right)} \underbrace{\mathbb{E}_{q\left(\boldsymbol{y}_{1:T}^s \mid \boldsymbol{z}^s, \boldsymbol{y}_0^s\right)}\left[\log \frac{p_\theta\left(\boldsymbol{y}_{0:T}^s \mid \boldsymbol{z}^s\right)}{q\left(\boldsymbol{y}_{1:T}^s \mid \boldsymbol{y}_0^s, \boldsymbol{z}^s\right)}\right]}_{\mathcal{L}_{ELBO}^d\left(\boldsymbol{z}^s, \boldsymbol{y}_0^s\right)} := \mathcal{L}_{ELBO}^s.
\end{aligned}
\tag{10}
$$

Eq. (10) implies a standard VAE and a classification diffusion model in the latent space. We provide the detailed derivation in Appendix A.1. To ensure discriminative classification boundaries in $\mathcal{Z}$, we additionally incorporate a deterministic classifier with parameter $\omega$ in the training objective. Then the overall objective for the labeled source data can be expressed as:

$$\mathcal{L}_s = -\mathcal{L}^s_{ELBO} + \mathbb{E}_{(\boldsymbol{x}^s, \boldsymbol{y}^s_0) \sim \mathcal{S}} \mathbb{E}_{q_\rho(\boldsymbol{z}^s | \boldsymbol{x}^s)} [-\log p_\omega(\boldsymbol{y}^s_0 \mid \boldsymbol{z}^s)]. \tag{11}$$

It is worth noting that the parameters of $f_\Omega$ have been further decomposed into $\{\rho, \omega\}$ in our case.

**Objective for Labeled Target Samples.** For target samples with annotation, we treat them the same as source-labeled data in training. The only difference is that we use parameter $\psi$ for target-specific reconstruction. Limited by space, we give the full expression of the training loss $\mathcal{L}^l_t$ in Appendix A.2.

**Objective for Unlabeled Target Samples.** The training objective for unlabeled target samples can not be carried out in a supervised fashion due to the absence of ground-truth labels. Therefore, we design the following variational lower bound for them by treating $\boldsymbol{y}^t_0$ and $\boldsymbol{z}^t$ as latent variables:

$$
\begin{aligned}
\log p\left(\boldsymbol{x}^t\right) &= \log \int p\left(\boldsymbol{x}^t, \boldsymbol{y}^t_0, \boldsymbol{z}^t\right) d\boldsymbol{y}^t_0 d\boldsymbol{z}^t \\
&\geq \mathbb{E}_{q\left(\boldsymbol{y}^t_0, \boldsymbol{z}^t | \boldsymbol{x}^t\right)} \left[\log \frac{p\left(\boldsymbol{x}^t, \boldsymbol{y}^t_0, \boldsymbol{z}^t\right)}{q\left(\boldsymbol{y}^t_0, \boldsymbol{z}^t \mid \boldsymbol{x}^t\right)}\right] \\
&= \underbrace{\mathbb{E}_{q_\rho(\boldsymbol{z}^t | \boldsymbol{x}^t)}[\log p_\psi(\boldsymbol{x}^t \mid \boldsymbol{z}^t)] - D_{KL}\left(q_\rho(\boldsymbol{z}^t \mid \boldsymbol{x}^t) \| p(\boldsymbol{z}^t)\right)}_{\text{VAE}} - \\
&\quad \underbrace{\mathbb{E}_{\boldsymbol{z}^t \sim q_\rho(\boldsymbol{z}^t | \boldsymbol{x}^t)}\left[D_{KL}(q(\boldsymbol{y}^t_0 \mid \boldsymbol{x}^t) \| p_\omega(\boldsymbol{y}^t_0 \mid \boldsymbol{z}^t)\right]}_{\mathcal{L}_{KD}} := \mathcal{L}^u_{ELBO}.
\end{aligned}
\tag{12}
$$

Intuitively, it encourages to optimize a standard VAE and minimize the KL divergence between $q(\boldsymbol{y}^t_0 \mid \boldsymbol{x}^t)$ and the approximation $p_\omega(\boldsymbol{y}^t_0 \mid \boldsymbol{z}^t)$. However, the ground-truth $q(\boldsymbol{y}^t_0 \mid \boldsymbol{x}^t)$ is agnostic in practice. In this work, we implement it in a Knowledge Distillation (KD) framework [53], where $q(\boldsymbol{y}^t_0 \mid \boldsymbol{x}^t)$ is substituted with the output of a teacher model $f_{\Omega'}$ that is optimized by exponential moving average (EMA) from the weights of student model $f_\Omega$. We denote the KD loss as $\mathcal{L}_{KD}$ and weight it by $\lambda_{kd}$ to absorb the influence of wrong predictions from the teacher model. And only those with confident predictions (larger than a pre-defined threshold $\alpha$) from the teacher model will be involved in this term. Meanwhile, we hope that the predictions of target samples to be individually confident and holistically diverse, which results in the following loss function:

$$\mathcal{L}^u_t = -\mathcal{L}^u_{ELBO} + \lambda_{kl} \underbrace{[D_{KL}(\mathbb{E}_{\boldsymbol{x}^t \sim \mathcal{T}_u} \mathbb{E}_{q_\rho(\boldsymbol{z}^t | \boldsymbol{x}^t)}[p_\omega(\boldsymbol{y}^t_0 \mid \boldsymbol{z}^t)] \| \frac{\mathbf{1}^K}{K}) - \mathbb{E}_{\boldsymbol{x}^t \sim \mathcal{T}_u} \mathbb{E}_{q_\rho(\boldsymbol{z}^t | \boldsymbol{x}^t)}[D_{KL}(p_\omega(\boldsymbol{y}^t_0 \mid \boldsymbol{z}^t) \| \frac{\mathbf{1}^K}{K})]]}_{\mathcal{L}_{KL}},$$

$$\tag{13}$$

where $\lambda_{kl}$ is the weighting coefficient. The derivation of Eq. (12) can be found in Appendix A.3.

**Adversarial Learning between Labeled and Unlabeled data.** The distribution shift between labeled and unlabeled data is a key obstacle to active learning. To extend the applicability of the diffusion model trained on labeled data to unlabeled ones, we apply a discriminator $D_\tau$ parameterized by $\tau$ and conduct adversarial learning between labeled and unlabeled data, expecting to learn an invariant latent space. Inspired by CDAN [22], we also leverage the model prediction as the conditioning for joint distribution alignment, which gives the following training loss for the discriminator:

$$\mathcal{L}_{adv} = -\mathbb{E}_{\boldsymbol{x} \sim (\mathcal{S} \cup \mathcal{T}_l)} \mathbb{E}_{\boldsymbol{z} \sim q_\rho(\boldsymbol{z} | \boldsymbol{x})} \log[D_\tau(p_\omega(\boldsymbol{y}_0 \mid \boldsymbol{z}), \boldsymbol{z})] - \mathbb{E}_{\boldsymbol{x} \sim \mathcal{T}_u} \mathbb{E}_{\boldsymbol{z} \sim q_\rho(\boldsymbol{z} | \boldsymbol{x})} \log[1 - D_\tau(p_\omega(\boldsymbol{y}_0 \mid \boldsymbol{z}), \boldsymbol{z})].$$

This minimax process is trained in an end-to-end mode using the gradient reversal layer [1].

### 3.3 Two-Stage Training Procedure

The whole model in our framework includes an encoder $\rho$, two decoders $\phi, \psi$, a discriminator $\tau$, a deterministic classifier $\omega$ and a diffusion-based classifier $\theta$. As shown in Section 3.1, the diffusion model requires the i.i.d. assumption to hold and injects the mean estimate $f_\Omega(\boldsymbol{x})$ into both forward and reverse diffusion process. Therefore, we schedule a two-stage training procedure in each active round. Specifically, the first stage is called the **adaptation stage**, which yields the following objective:

$$\max_\tau \min_{\rho, \phi, \psi, \omega} \mathcal{L}_s + \mathcal{L}^u_t + \mathcal{L}^l_t - \mathcal{L}_{adv}. \tag{14}$$

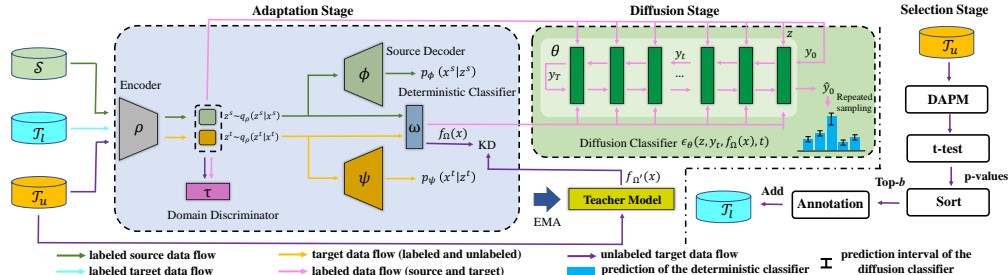

Figure 1: Framework of DAPM. Each active learning round goes through three stages: two stages for training and one stage for selection. Parameters in blue and green backgrounds are trained in the adaptation stage and the diffusion stage, respectively. When the adaptation stage ends, the parameters involved will be frozen. The teacher model $f_{\Omega'}$ is updated by EMA from the student model $f_\Omega$.

The first stage ensures a well-behaved mean estimator $f_\Omega$ and a domain-invariant latent embedding. We then freeze all the trained parameters and train the diffusion model $\epsilon_\theta (z, y_t, f_\Omega(x), t)$ in the second stage, namely the **diffusion stage**, with all labeled data:

$$\min_\theta \mathcal{L}_s + \mathcal{L}_t^l. \tag{15}$$

Having the diffusion model, we select target samples according to the strategy in Section 3.4 if there is a remaining budget. We depict the whole framework for ADA in Fig. 1.

**Extension to Source-Free ADA (SFADA).** We naturally extend our method to SFADA [47] based on the fact that the model $f_\Omega := \{\rho, \omega\}$ in the adaptation stage can be initialized (like in ADA) or source pre-trained. For the latter, we pre-train the source model with $\mathcal{L}_s$, and disable the source-related losses, i.e., $\mathcal{L}_s$ and $\mathcal{L}_{adv}$ in the adaptation stage. Besides, we use confident unlabeled target data pseudo-labeled by the teacher model to substitute the source samples in $\mathcal{L}_s$ in the diffusion stage. The detailed procedure of the training process can be found in Appendix. B.

## 3.4 T-test-Based Selection Strategy

Existing ADA methods generally select both target-representative and uncertain samples. The former characteristic can be naturally absorbed within the adaptation stage, which makes our model operate in a domain-agnostic latent space. Therefore, we mainly focus on selecting samples with high predictive uncertainties. For each $x^t \sim \mathcal{T}_u$, we generate $N$ predictions independently with the diffusion classifier, denoted by $\{\widehat{y}_n^t\}_{n=1}^N$. Since the raw predictions can be arbitrary real vectors, we then convert them into the probability simplex by a softmax operation following [19], resulting in a set $\{\widetilde{y}_n^t\}_{n=1}^N$. Based on that, we identify the two most predicted classes among $N$ votes for each instance, which gathers two groups of predicted scores $g_1 = \{\widetilde{y}_n^t[a]\}_{n=1}^N$ and $g_2 = \{\widetilde{y}_n^t[b]\}_{n=1}^N$, where $a, b$ denote two most voted class labels. We assume independent generation in each dimension and evaluate the uncertainty by conducting an independent two-sample t-test, where the t-value is:

$$t = \frac{\bar{g}_1 - \bar{g}_2}{S_{\bar{g}_1 - \bar{g}_2}}, S_{\bar{g}_1 - \bar{g}_2} = \sqrt{\frac{s_1^2 + s_2^2}{N}}, \tag{16}$$

here we use $\bar{g}_1$ (or $\bar{g}_2$) and $s_1^2$ (or $s_2^2$) to denote the mean and variance of the two groups, respectively. Conceptually, the t-value is formulated as the ratio of *mean difference between groups* to *sampling variability*. Compared to Best-versus-Second-Best (BvSB) [36] used in deterministic model-based AL methods, the t-test-based criterion can not only allow us to see if two means are different, but also tell us how significant the differences are. As a result, we use the p-value to express the significance of such a difference. Intuitively, uncertain samples are those with small mean differences between groups and high sampling variabilities, which leads to small t-values and high p-values. In practice, we choose unlabeled target samples with top-$b$ p-values for annotation in each active selection round.

## 4 Experiments

### 4.1 Experimental Setup

**Datasets and Baselines.** We evaluate our method on three widely used domain adaptation benchmarks, i.e., Office-31 [54], Office-Home [55] and VisDA [56]. We construct three groups of baselines

Table 1: Accuracy (%) on VisDA and Office-31 datasets under different settings with 5% labeled target samples (ResNet-50).

| Category | Method | VisDA | Office-31 | | | | | | |
|---|---|---|---|---|---|---|---|---|---|
| | | Synthetic $\rightarrow$ Real | A $\rightarrow$ D | A $\rightarrow$ W | D $\rightarrow$ A | D $\rightarrow$ W | W $\rightarrow$ A | W $\rightarrow$ D | Avg |
| Source-Only | ResNet [58] | $44.7 \pm 0.1$ | 81.5 | 75.0 | 63.1 | 95.2 | 65.7 | 99.4 | 80.0 |
| Active Learning | Random | $78.6 \pm 0.6$ | 87.1 | 84.1 | 75.5 | 98.1 | 75.8 | 99.6 | 86.7 |
| | BvSB [36] | $81.3 \pm 0.4$ | 89.8 | 87.9 | 78.2 | 99.0 | 78.6 | **100.0** | 88.9 |
| | Entropy [7] | $82.7 \pm 0.3$ | 91.0 | 89.2 | 76.1 | 99.7 | 77.7 | **100.0** | 88.9 |
| | CoreSet [33] | $81.9 \pm 0.3$ | 82.5 | 81.1 | 70.3 | 96.5 | 72.4 | 99.6 | 83.7 |
| | BADGE [57] | $84.3 \pm 0.3$ | 90.8 | 89.1 | 79.8 | 99.6 | 79.6 | **100.0** | 89.8 |
| Active DA | AADA [37] | $80.8 \pm 0.4$ | 89.2 | 87.3 | 78.2 | 99.5 | 78.7 | **100.0** | 88.8 |
| | TQS [11] | $83.1 \pm 0.4$ | 92.8 | 92.2 | 80.6 | **100.0** | 80.4 | **100.0** | 91.1 |
| | CLUE [4] | $85.2 \pm 0.4$ | 92.0 | 87.3 | 79.0 | 99.2 | 79.6 | 99.8 | 89.5 |
| | EADA [5] | $88.3 \pm 0.1$ | 97.7 | 96.6 | 82.1 | 100.0 | 82.8 | **100.0** | 93.2 |
| | * DUC [20] | $88.9 \pm 0.2$ | 95.8 | 96.4 | 81.9 | 99.6 | 81.4 | **100.0** | 92.5 |
| | DAPM (Baseline) | $80.8 \pm 0.7$ | 94.3 | 93.5 | 72.1 | 97.6 | 72.3 | 99.3 | 88.2 |
| | DAPM-TT | $\mathbf{89.1 \pm 0.1}$ | 96.8 | **98.6** | 82.3 | 99.8 | **83.3** | **100.0** | **93.5** |
| SFADA | * ELPT [47] | $83.5 \pm 0.6$ | **98.0** | 97.2 | 81.2 | 99.4 | 80.7 | **100.0** | 92.8 |
| | DAPM (Baseline) | $73.4 \pm 0.2$ | 93.7 | 92.9 | 72.2 | 98.1 | 72.6 | 99.1 | 88.1 |
| | † Random | $86.2 \pm 0.8$ | 94.4 | 95.1 | 78.2 | 98.1 | 79.2 | 99.6 | 90.8 |
| | † BvSB [36] | $88.1 \pm 0.2$ | 96.7 | 96.5 | 81.1 | 99.0 | 81.5 | **100.0** | 92.5 |
| | † Entropy [7] | $87.5 \pm 0.3$ | 96.3 | 95.6 | 80.3 | 99.8 | 80.5 | **100.0** | 91.9 |
| | † CoreSet [33] | $87.3 \pm 0.1$ | 94.7 | 96.5 | 80.0 | 97.9 | 79.9 | **100.0** | 91.5 |
| | † BADGE [57] | $87.9 \pm 0.3$ | 96.5 | 97.1 | 79.3 | 98.5 | 80.1 | 99.8 | 91.9 |
| | DAPM-TT | $88.4 \pm 0.3$ | 96.8 | 96.4 | **83.5** | 99.7 | 81.7 | **100.0** | 93.0 |

\* For DUC and ELPT, we report the results on Office-31 and VisDA based on our own runs, respectively, according to the official code.
† For SFADA, we implement several active learning methods upon our SFDA baseline and report the mean results over 3 runs.

for comparison. (i) Active learning: Random, Entropy [7], BvSB [36], CoreSet [33], BADGE [57]. (ii) Active Domain Adaptation: AADA [37], TQS [11], CLUE [4], EADA [5], DUC [20]. (iii) Source-Free ADA: ELPT [47], and our SFDA baseline with other AL selection methods: Random (DAPM-RD), BvSB (DAPM-BS), Entropy (DAPM-ET), CoreSet (DAPM-CS), BADGE (DAPM-BG). We denote our method with the t-test-based selection criterion by DAPM-TT thereafter. It is worth noting that in the SFADA setting, we retain the abbreviation as DAPM for consistency, however we do not employ the adversarial learning loss.

**Implementation.** We implement our method on Pytorch and MindSpore[3]. The ResNet-50 [58] pre-trained on ImageNet [59] is adopted as the backbone, which constitutes the main body of the encoder. The decoder is simply a two-layer MLP. Following previous works [5, 20], we schedule 5 selection rounds, where in each round, the model selects $b = 1\% \times n_t$ samples for annotation, and therefore the total budget $B = 5\% \times n_t$. We use SGD optimizer in the adaptation stage. The learning rate is set as 0.01 except for the VisDA dataset in SFADA, where we use 0.001 for training. The batch size is 32 for ADA and 64 for SFADA. We adopt the same learning rate scheduler as [22, 1]. For hyperparameters, we use $\lambda_{kl} = 0.1$, $\lambda_{kd} = 0.1$ for ADA and $\lambda_{kl} = 1.0$, $\lambda_{kd} = 1.0$ for SFADA. For all tasks, the confidence threshold $\alpha$ for knowledge distillation is set to 0.9. We set $N = 100$, i.e., we generate 100 predictions for each individual sample for uncertainty estimation. More implementation details are provided in Appendix. C. Code is available at `https://github.com/TL-UESTC/DAPM`.

**Evaluation Protocol.** There are two classifiers involved in our method, i.e., a deterministic classifier parameterized by $\omega$ and a diffusion classifier parameterized by $\theta$. For evaluating a target sample $\boldsymbol{x}^t$ encoded by $\boldsymbol{z}^t$, we leverage the predictions generated from the diffusion classifier by calculating the expected class probability based on $N$ independent votes, i.e., $\bar{p}_\theta(\boldsymbol{y}^t \mid \boldsymbol{z}^t) = \frac{1}{N} \sum_{n=1}^{N} \widetilde{\boldsymbol{y}}_n^t$. We show in Sec. 4.3 that $\bar{p}_\theta(\boldsymbol{y}^t \mid \boldsymbol{z}^t)$ provides a more calibrated prediction than the output of the deterministic classifier $p_\omega(\boldsymbol{y}^t \mid \boldsymbol{z}^t)$. On the other hand, we use the majority vote as the final predicted class.

## 4.2 Main Results

**Active Domain Adaptation.** We present our results on Office-31 and VisDA in Table 1. Notably, among traditional active selection strategies, those based on uncertainty (e.g., BvSB, Entropy, BADGE) tend to achieve higher performance compared to those based purely on representativeness (e.g., CoreSet). This is because a portion of the target domain is already well-aligned with the source domain, and selecting these easy samples would not provide informative feedback to the model. This highlights the importance of incorporating predictive uncertainty in the selection criterion. However, the accuracies of all traditional AL methods are significantly lower than those of ADA methods due

---
[3]https://www.mindspore.cn/

Table 2: Accuracy (%) on Office-Home dataset under different settings with 5% labeled target samples (ResNet-50). Marks have the same meaning as in Table 1.

| Category | Method | Ar→Cl | Ar→Pr | Al→Rw | Cl→Ar | Cl→Pr | Cl→Rw | Pr→Ar | Pr→Cl | Pr→Rw | Rw→Ar | Rw→Cl | Rw→Pr | Avg |
|---|---|---|---|---|---|---|---|---|---|---|---|---|---|---|
| Source-Only | ResNet | 42.1 | 66.3 | 73.3 | 50.7 | 59.0 | 62.6 | 51.9 | 37.9 | 71.2 | 65.2 | 42.6 | 76.6 | 58.3 |
| Active Learning | Random | 52.5 | 74.3 | 77.4 | 56.3 | 69.7 | 68.9 | 57.7 | 50.9 | 75.8 | 70.0 | 54.6 | 81.3 | 65.8 |
| | BvSB [36] | 56.3 | 78.6 | 79.3 | 58.1 | 74.0 | 70.9 | 59.5 | 52.6 | 77.2 | 71.2 | 56.4 | 84.5 | 68.2 |
| | Entropy [7] | 58.0 | 78.4 | 79.1 | 60.5 | 73.0 | 72.6 | 60.4 | 54.2 | 77.9 | 71.3 | 58.0 | 83.6 | 68.9 |
| | CoreSet [33] | 51.8 | 72.6 | 75.9 | 58.3 | 68.5 | 70.1 | 58.8 | 48.8 | 75.2 | 69.0 | 52.7 | 80.0 | 65.1 |
| | BADGE [57] | 58.2 | 79.7 | 79.9 | 61.5 | 74.6 | 72.9 | 61.5 | 56.0 | 78.3 | 71.4 | 60.9 | 84.2 | 69.9 |
| Active DA | AADA [37] | 56.6 | 78.1 | 79.0 | 58.5 | 73.7 | 71.0 | 60.1 | 53.1 | 77.0 | 70.6 | 57.0 | 84.5 | 68.3 |
| | TQS [11] | 58.6 | 81.1 | 81.5 | 61.1 | 76.1 | 73.3 | 61.2 | 54.7 | 79.7 | 73.4 | 58.9 | 86.1 | 70.5 |
| | CLUE [4] | 58.0 | 79.3 | 80.9 | 68.8 | 77.5 | 76.7 | 66.3 | 57.9 | 81.4 | 75.6 | 60.8 | 86.3 | 72.5 |
| | EADA [5] | 63.6 | 84.4 | 83.5 | 70.7 | 83.7 | 80.5 | 73.0 | 63.5 | 85.2 | 78.4 | 65.4 | 88.6 | 76.7 |
| | DUC [20] | **65.5** | 84.9 | 84.3 | **73.0** | 83.4 | 81.1 | **73.9** | **66.6** | 85.4 | **80.1** | **69.2** | 88.8 | **78.0** |
| | DAPM (Baseline) | 51.9 | 73.1 | 78.4 | 59.6 | 76.3 | 74.9 | 61.2 | 53.5 | 80.2 | 70.2 | 58.3 | 81.3 | 68.2 |
| | DAPM-TT | 64.2 | 85.4 | **85.7** | 69.2 | 84.2 | 83.5 | 69.1 | 63.4 | **86.0** | 77.2 | 68.4 | 88.6 | 77.1 |
| SFADA | ELPT [47] | 65.3 | 84.1 | 84.9 | 72.9 | 84.4 | 82.8 | 69.8 | 63.3 | 86.1 | 76.2 | 65.6 | **89.1** | 77.0 |
| | DAPM (Baseline) | 50.5 | 76.1 | 80.6 | 66.5 | 74.9 | 77.8 | 63.8 | 49.9 | 80.1 | 72.4 | 53.9 | 83.5 | 69.2 |
| | † Random | 58.5 | 82.4 | 82.3 | 68.8 | 81.0 | 80.6 | 69.4 | 60.5 | 82.2 | 76.2 | 64.2 | 85.6 | 74.3 |
| | † BvSB [36] | 62.5 | 83.5 | 83.4 | 72.1 | 84.5 | 83.0 | 70.3 | 60.4 | 85.5 | 76.3 | 63.7 | 87.4 | 76.1 |
| | † Entropy [7] | 59.2 | 82.1 | 84.2 | 68.2 | 82.0 | 80.2 | 66.2 | 58.7 | 84.1 | 75.7 | 63.3 | 87.6 | 74.3 |
| | † CoreSet [33] | 61.2 | 83.5 | 85.0 | 70.7 | 82.5 | 82.6 | 68.9 | 60.3 | 83.5 | 76.3 | 63.8 | 86.9 | 75.4 |
| | † BADGE [57] | 60.2 | 83.9 | 84.9 | 71.8 | 83.7 | 81.6 | 69.1 | 59.8 | 85.1 | 75.9 | 62.8 | 88.1 | 75.6 |
| | DAPM-TT | 64.4 | **85.8** | 85.4 | 72.4 | **84.7** | **84.1** | 70.0 | 63.3 | 85.6 | 77.4 | 65.8 | **89.1** | 77.3 |

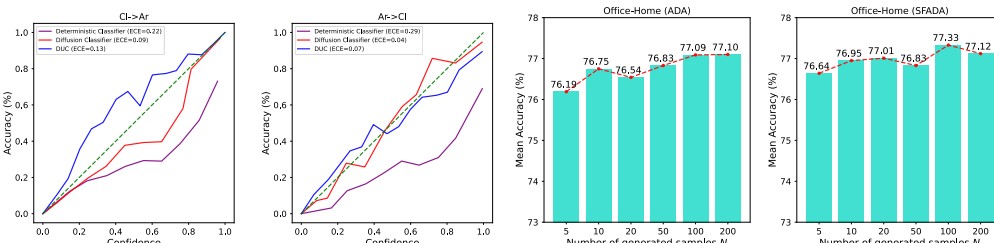

(a) ECE of the target data on Office-Home.  (b) Performance under different $N$ on Office-Home.

Figure 2: (a) Visualization of Expected Calibration Error (ECE) for ADA on task Ar → Cl and Cl → Ar. (b) Performance under different choices of $N$ on Office-Home for ADA and SFADA.

to the neglect of domain shift. Among ADA methods, our DAPM-TT outperforms other methods and achieves the best accuracies on both Office-31 and VisDA, surpassing the deterministic model EADA [5] by 0.8% and 0.3% on the two datasets, respectively. Compared to the recent method DUC [20], our method still achieves better performance, boosting the accuracy by 1.0% on Office-31 and achieving a slightly better performance on VisDA, showing the superiority of our probablistic framework. The results in Table 2 demonstrate that our method outperforms all deterministic ADA methods on the Office-Home dataset. Although it does not achieve the best accuracy in this case, it remains highly competitive with DUC. Moreover, it is worth noting that our probabilistic modeling is mainly designed to capture predictive uncertainty, rather than to optimize classification accuracy that is largely depended on domain adaptation methods. We will show in Sec. 4.3 that our method induces more calibrated predictions than DUC.

**Source-Free Active Domain Adaptation.** For SFADA, we mainly compare our DAPM-TT with other AL strategies using our SFDA baseline, where we conduct the adaptation stage and adopt some representative AL strategies based on the deterministic classifier. The results in Table 1 and Table 2 show that DAPM-TT consistently achieves the best performance on all datasets. In particular, we find that BvSB [36], which can be considered as the deterministic counterpart of DAPM-TT, achieves relatively good results on all datasets. However, our DAPM-TT outperforms BvSB by considering the full distribution of the predicted variable, providing a better measure of uncertainty. Compared to the recent SFADA method ELPT [47], our DAPM-TT is able to surpass it on all datasets, especially the VisDA dataset, where we achieve a 4.9% improvement in mean accuracy.

### 4.3 Analytical Experiments

**Expected Calibration Error (ECE).** To evaluate the calibration ability of our model, we plot the ECE curves for two tasks on Office-Home in Fig. 2a. The results demonstrate that, in both tasks, the output of our deterministic classifier is much more uncalibrated compared to that of the diffusion classifier, even though their accuracies remain very close (as shown in the Ablation Study). This confirms the well-known phenomenon that softmax classifiers based on point estimates tend to make overconfident predictions [60], which can be unsafe when dealing with uncertain estimation. Our DAPM aims to recover the full predictive distribution by modeling the uncertainty in both data

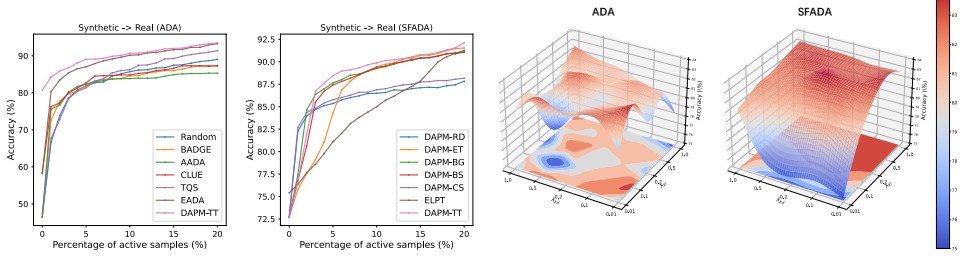

(a) Accuracies under different budgets on VisDA.    (b) Parameter sensitivity analysis on task D → A.

Figure 3: (a) Accuracy curves as budget goes from 0% to 20%. (b) Model performance varying with parameters $\lambda_{kl}, \lambda_{kd} \in \{0.01, 0.1, 0.2, 0.5, 1.0\}$ on task D → A (Office-31).

Table 3: Comparasion between different selection strategies for ADA (5% budget on Office-31).

| Method | Deterministic Classifier | | | | | Diffusion Classifier | | | |
|---|---|---|---|---|---|---|---|---|---|
| | DAPM-RD | DAPM-BS | DAPM-ET | DAPM-CS | DAPM-BG | DAPM-VT | DAPM-IW | DAPM-AE | DAPM-TT |
| Accuracy (%) | 92.0 | 92.2 | 92.1 | 91.8 | 92.5 | 93.1 | 92.9 | 92.7 | **93.5** |

generation and model prediction, thereby effectively mitigating this issue. Compared to DUC [20], which only models the distribution in output space, our method achieves smaller ECE values in both tasks, demonstrating the benefits of our probabilistic framework.

**Effect of Different Sampling Scales.** We evaluate our model under varying sampling numbers $N$ and report the performance on Office-Home in Fig. 2b. Our results demonstrate that the model performance initially improves as $N$ increases. This is because when $N$ is too small, the model suffers from sampling bias, which results in a performance degradation to a level similar to that of the point estimate. However, an excessively large number of samples can lead to increased storage and computing resources. We find that when $N$ is around 100, the performance becomes stable.

**Effect of Different Annotation Budgets.** Fig. 3a shows how different budgets affect the model performance. For both ADA and SFADA, our method is able to select the most informative samples at the begining, resulting in superior performance even with a small budget. As the budget increases, the advantage of our method is slightly diluted. However, it is still able to maintain a leading accuracy.

**Evaluation of Active Learning.** We also provide the baseline domain adaptation performance (i.e., DAPM without active learning) in Table 1 and 2. It shows that without active selection and learning, our DAPM only achieves a moderate performance compared to modern methods that are specifically designed for UDA, given that the probabilistic backbone is mainly used for uncertainty modeling rather than accuracy. However, we are encouraged to observe that incorporating AL with our designed selection criterion significantly enhances the domain adaptation performance on all datasets. This outcome demonstrates the effectiveness of our diffusion-based uncertainty estimation, as it identifies informative samples that considerably improve overall performance.

**Performance under Different Selection Strategies.** To explore the superiority of our t-test-based selection strategy, we test our DAPM using other selection strategies for the ADA task. The results are presented in Table 3. In addition to strategies based on the deterministic classifier, we also evaluate three strategies based on the diffusion classifier: DAPM-VT, DAPM-IW and DAPM-AE. DAPM-VT, DAPM-IW select samples with the top-$B$ highest prediction variance and interval width on the majority voted class, respectively. DAPM-AE averages the entropy across the $N$ predictions to get the final nonparametric entropy estimate for each data point, and select the ones with highest averaged entropies for annotation. As expected, DAPM achieved the best results. Our findings suggest that the t-test-based criterion is more suitable for the diffusion classifier since it takes into account both sampling variability and cross-category ambiguity.

**T-SNE Visualization of Latent Representations.** We visualize the latent representations of unlabeled target data and selected target data in Fig. 4 using t-SNE [61]. In this visualization experiment, we compared our DAPM-TT with BvSB [36] that is based on the deterministic classifier. It can be observed that BvSB tends to select samples from relatively ambiguous regions (the center region) since these samples often have ambiguity between different classes. However, many samples selected by BvSB are in areas where the model is able to make predictions accurately. Therefore, it will not help to correct the samples with wrong predictions, resulting in modest improvement on performance.

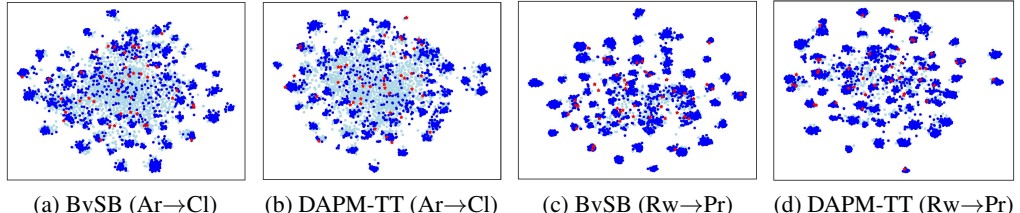

| (a) BvSB (Ar→Cl) | (b) DAPM-TT (Ar→Cl) | (c) BvSB (Rw→Pr) | (d) DAPM-TT (Rw→Pr) |

Figure 4: Visualization of latent representations using t-SNE [61] on task Ar→Cl (a to b) and Rw→Pr (c to d) of ADA. Darkblue points are unlabeled target samples correctly classified by our model. Lightblue points represent unlabeled target samples misclassified by our model. Red stars are the selected target samples.

Our DAMP-TT, on the other hand, can select samples from both the regions where there is ambiguity between classes and the regions where a large number of samples are misclassified, which are exaclty the ones we want to select for annotation.

**How Does T-test-based Selection Ensure Diversity?** While our method mainly focuses on uncertainty estimation, we show that it has properties that help mitigate redundant selections. Firstly, the diffusion process introduces stochasticity by generating varied predictions for the same input. This creates more diversity in the uncertainty estimates across similar instances. In other words, duplicate inputs will not necessarily have identical uncertainty. Secondly, the t-test criterion accounts for both variability across predictions and similarity of top-2 classes. Highly variable samples with closer competing classes will be favored, which is naturally distributed around the classification boundary of every class, thus enabling diversity. In Fig. 4, we observe that our approach naturally selects a diverse range of sample classes, even though we did not explicitly impose a diversity constraint.

**Ablation Study.** We conduct extensive experiments to investigate the influence of different classifiers and losses in our method. The results are summarized in Table 4, and we observed the following: Firstly, in most cases, the diffusion classifier yields slightly better results than the deterministic one, indicating its superior tolerance for domain shift. Secondly, our approach still achieves much better results than other traditional AL methods without any loss function for domain adaptation, demonstrating the superiority of our selection method. Lastly, all the loss functions have a positive effect on the final result.

**Hyperparameter Sensitivity.** As shown in Fig. 3b, our model exhibits low sensitivity to both $\lambda_{kl}$ and $\lambda_{kd}$ in ADA, and moderate sensitivity to $\lambda_{kl}$ in SFADA. We conjecture the reason is that the source-available scenario involves more loss terms in the training objective, the additional source domain data and distribution alignment objectives like $\mathcal{L}_{adv}$ may make the optimization landscape more complex and susceptible to suboptimal solutions based on weighting hyperparameters. In contrast, in SFADA, the model solely relies on the target data and regularization losses like $\mathcal{L}_{KL}$ for alignment, reducing dependence on precise weighting. As SFADA lacks source-supervised information, we recommend using a slightly larger $\lambda_{kl}$ to ensure good adaptation performance.

Table 4: Ablation study of DAPM-TT on Office-31 and VisDA. Dif. and Det. are short for diffusion classifier and deterministic classifier, respectively.

| Classifier | | Loss | | | Office-31 | | VisDA-2017 | |
|---|---|---|---|---|---|---|---|---|
| Dif. | Det. | $\mathcal{L}_{KD}$ | $\mathcal{L}_{KL}$ | $\mathcal{L}_{adv}$ | ADA | SFADA | ADA | SFADA |
| ✓ | | | | | 91.4 | 90.8 | 86.3 | 86.7 |
| | ✓ | | | | 91.1 | 91.0 | 86.5 | 86.5 |
| ✓ | | ✓ | | | 92.2 | 91.9 | 87.1 | 87.6 |
| ✓ | | ✓ | | ✓ | 92.9 | - | 87.7 | - |
| ✓ | | ✓ | ✓ | | 93.0 | 93.0 | 88.5 | 88.4 |
| | ✓ | ✓ | ✓ | | 93.0 | 93.1 | 88.3 | 88.2 |
| ✓ | | ✓ | ✓ | ✓ | **93.5** | - | **89.1** | - |
| | ✓ | ✓ | ✓ | ✓ | 93.4 | - | 88.9 | - |

## 5 Conclusion

In this work, we propose a novel probabilistic framework for ADA that leverages the variability of both latent data representation and model prediction for better uncertainty estimation. Our approach combines a variational autoencoder, a diffusion probabilistic classifier, and an auxiliary deterministic classifier to guide training and ensure an invariant latent space. Our experiments on three domain adaptation benchmarks demonstrate the effectiveness of our approach in improving task performance and effectively handling uncertainty estimation for both ADA and SFADA.

## Acknowledgments

This work was supported in part by the National Natural Science Foundation of China under Grant 62173066, 62250061 and 62176042, and in part by Sichuan Science and Technology Program under Grant 2023NSFSC0483, and in part Sponsored by CAAI-Huawei MindSpore Open Fund.

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

## Appendix Contents

## A Derivations

### A.1 Derivation of the Labeled Source Objective

Given a labeled source sample $(\boldsymbol{x}^s, \boldsymbol{y}_0^s)$, our goal is to inference the latent data representation $\boldsymbol{z}^s$ and a sequence of latent class representations $\boldsymbol{y}_{1:T}^s$, which controls the generation process of data points and predictions, respectively. The log-likelihood of the labeled source data can thus be expressed by:

$$\log p\left(\boldsymbol{x}^s, \boldsymbol{y}_0^s\right) = \log \int p\left(\boldsymbol{x}^s, \boldsymbol{y}_{0:T}^s, \boldsymbol{z}^s\right) d\boldsymbol{y}_{1:T}^s d\boldsymbol{z}^s. \tag{17}$$

Since Eq. (17) is intractable to compute in practice, we then leverage variational inference to approximate the posterior distribution of unknown variables $(\boldsymbol{z}^s, \boldsymbol{y}_{1:T}^s)$ and solve it by optimizing the following ELBO:

$$\log \int p\left(\boldsymbol{x}^s, \boldsymbol{y}_{0:T}^s, \boldsymbol{z}^s\right) d\boldsymbol{y}_{1:T}^s d\boldsymbol{z}^s \geq \mathbb{E}_{q\left(\boldsymbol{y}_{1:T}^s, \boldsymbol{z}^s \mid \boldsymbol{x}^s, \boldsymbol{y}_0^s\right)}\left[\log \frac{p\left(\boldsymbol{y}_{0:T}^s, \boldsymbol{x}^s, \boldsymbol{z}^s\right)}{q\left(\boldsymbol{y}_{1:T}^s, \boldsymbol{z}^s \mid \boldsymbol{x}^s, \boldsymbol{y}_0^s\right)}\right], \tag{18}$$

where $q(\boldsymbol{z}^s, \boldsymbol{y}_{1:T}^s \mid \boldsymbol{x}^s, \boldsymbol{y}_0^s)$ is the approximation of the ground-truth joint posterior $p(\boldsymbol{z}^s, \boldsymbol{y}_{1:T}^s \mid \boldsymbol{x}^s, \boldsymbol{y}_0^s)$, which can be further factorized as:

$$q(\boldsymbol{z}^s, \boldsymbol{y}_{1:T}^s \mid \boldsymbol{x}^s, \boldsymbol{y}_0^s) = q_\rho(\boldsymbol{z}^s \mid \boldsymbol{x}^s, \boldsymbol{y}_0^s) q(\boldsymbol{y}_{1:T}^s \mid \boldsymbol{y}_0, \boldsymbol{z}^s, \boldsymbol{x}^s). \tag{19}$$

With Eq. (19) and the generative process assumed in Eq. (8), we have the following derivation:

$$
\begin{aligned}
\log p\left(\boldsymbol{x}^s, \boldsymbol{y}_0^s\right) &= \log \int p\left(\boldsymbol{x}^s, \boldsymbol{y}_{0:T}^s, \boldsymbol{z}^s\right) d\boldsymbol{y}_{1:T}^s d\boldsymbol{z}^s \\
&\geq \mathbb{E}_{q\left(\boldsymbol{y}_{1:T}^s, \boldsymbol{z}^s \mid \boldsymbol{x}^s, \boldsymbol{y}_0^s\right)}\left[\log \frac{p\left(\boldsymbol{y}_{0:T}^s, \boldsymbol{x}^s, \boldsymbol{z}^s\right)}{q\left(\boldsymbol{y}_{1:T}^s, \boldsymbol{z}^s \mid \boldsymbol{x}^s, \boldsymbol{y}_0^s\right)}\right] \\
&= \mathbb{E}_{q\left(\boldsymbol{y}_{1:T}^s, \boldsymbol{z}^s \mid \boldsymbol{x}^s, \boldsymbol{y}_0^s\right)}\left[\log \frac{p(\boldsymbol{z}^s) p_\phi(\boldsymbol{x}^s \mid \boldsymbol{z}^s) p_\theta\left(\boldsymbol{y}_{0:T}^s \mid \boldsymbol{x}^s, \boldsymbol{z}^s\right)}{q_\rho(\boldsymbol{z}^s \mid \boldsymbol{x}^s, \boldsymbol{y}_0^s) q(\boldsymbol{y}_{1:T}^s \mid \boldsymbol{y}_0, \boldsymbol{z}^s, \boldsymbol{x}^s)}\right] \\
&= \mathbb{E}_{q\left(\boldsymbol{y}_{1:T}^s, \boldsymbol{z}^s \mid \boldsymbol{x}^s, \boldsymbol{y}_0^s\right)}\left[\log \frac{p(\boldsymbol{z}^s)}{q_\rho(\boldsymbol{z}^s \mid \boldsymbol{y}_0, \boldsymbol{x})} + \log p_\phi(\boldsymbol{x}^s \mid \boldsymbol{z}^s) + \log \frac{p_\theta\left(\boldsymbol{y}_{0:T}^s \mid \boldsymbol{x}^s, \boldsymbol{z}^s\right)}{q(\boldsymbol{y}_{1:T}^s \mid \boldsymbol{y}_0^s, \boldsymbol{z}^s, \boldsymbol{x}^s)}\right] \\
&= \mathbb{E}_{q_\rho\left(\boldsymbol{z}^s \mid \boldsymbol{x}^s, \boldsymbol{y}_0^s\right)}\left[\log \frac{p(\boldsymbol{z}^s)}{q_\rho(\boldsymbol{z}^s \mid \boldsymbol{y}_0^s, \boldsymbol{x}^s)} + \log p_\phi(\boldsymbol{x}^s \mid \boldsymbol{z}^s)\right] + \\
&\quad \mathbb{E}_{q\left(\boldsymbol{y}_{1:T}^s, \boldsymbol{z}^s \mid \boldsymbol{x}^s, \boldsymbol{y}_0^s\right)}\left[\log \frac{p_\theta\left(\boldsymbol{y}_{0:T}^s \mid \boldsymbol{x}^s, \boldsymbol{z}^s\right)}{q(\boldsymbol{y}_{1:T}^s \mid \boldsymbol{y}_0^s, \boldsymbol{z}^s, \boldsymbol{x}^s)}\right] \\
&= \mathbb{E}_{q_\rho\left(\boldsymbol{z}^s \mid \boldsymbol{x}^s, \boldsymbol{y}_0^s\right)}\left[\log p_\phi(\boldsymbol{x}^s \mid \boldsymbol{z}^s)\right] - D_{KL}\left(q_\rho\left(\boldsymbol{z}^s \mid \boldsymbol{x}^s, \boldsymbol{y}_0^s\right) \| p(\boldsymbol{z}^s)\right) + \\
&\quad \mathbb{E}_{\boldsymbol{z}^s \sim q_\rho\left(\boldsymbol{z}^s \mid \boldsymbol{x}^s, \boldsymbol{y}_0^s\right)} \underbrace{\mathbb{E}_{q\left(\boldsymbol{y}_{1:T}^s \mid \boldsymbol{x}^s, \boldsymbol{z}^s, \boldsymbol{y}_0^s\right)}\left[\frac{p_\theta\left(\boldsymbol{y}_{0:T}^s \mid \boldsymbol{x}^s, \boldsymbol{z}^s\right)}{q\left(\boldsymbol{y}_{1:T}^s \mid \boldsymbol{y}_0^s, \boldsymbol{z}^s, \boldsymbol{x}^s\right)}\right]}_{①}.
\end{aligned}
\tag{20}
$$

In ①, the forward diffusion process $q\left(\boldsymbol{y}_{1:T}^s \mid \boldsymbol{x}^s, \boldsymbol{z}^s, \boldsymbol{y}_0^s\right)$ and the reverse diffusion process $p_\theta\left(\boldsymbol{y}_{0:T}^s \mid \boldsymbol{x}^s, \boldsymbol{z}^s\right)$ are still based on the original input $\boldsymbol{x}^s$. In this work, we make a simplification design to assume that the observed class variable $\boldsymbol{y}_0^s$ and latent ones $\boldsymbol{y}_{1:T}^s$ are only conditioned

on the latent variable $z^s$. We demonstrate the reasonability of this simplification in Appendix A.3. Consequently, we have

$$① = \mathbb{E}_{q\left(\boldsymbol{y}_{1:T}^s | \boldsymbol{z}^s, \boldsymbol{y}_0^s\right)} \left[ \frac{p_\theta \left(\boldsymbol{y}_{0:T}^s \mid \boldsymbol{z}^s\right)}{q\left(\boldsymbol{y}_{1:T}^s \mid \boldsymbol{y}_0^s, \boldsymbol{z}^s\right)} \right], \tag{21}$$

which has the same form of ELBO as the diffusion classifier in Eq. (4).

## A.2 Expression of the Labeled Target Objective

For the labeled target data $(\boldsymbol{x}^t, \boldsymbol{y}_0^t)$, the unknown latent variables are the same as the labeled source data, and therefore the ELBO is analogous to that of the labeled source data. The only difference is that we use the target-specific decoder $p_\psi(\boldsymbol{x}^t \mid \boldsymbol{z}^t)$. We give the full expression as follows:

$$
\begin{aligned}
\log p\left(\boldsymbol{x}^t, \boldsymbol{y}_0^t\right) &= \log \int p\left(\boldsymbol{x}^t, \boldsymbol{y}_{0:T}^t, \boldsymbol{z}^t\right) d\boldsymbol{y}_{1:T}^t d\boldsymbol{z}^t \\
&\geq \mathbb{E}_{q\left(\boldsymbol{y}_{1:T}^t, \boldsymbol{z}^t | \boldsymbol{x}^t, \boldsymbol{y}_0^t\right)} \left[ \log \frac{p\left(\boldsymbol{y}_{0:T}^t, \boldsymbol{x}^t, \boldsymbol{z}^t\right)}{q\left(\boldsymbol{y}_{1:T}^t, \boldsymbol{z}^t \mid \boldsymbol{x}^t, \boldsymbol{y}_0^t\right)} \right] \\
&= \mathbb{E}_{q\left(\boldsymbol{y}_{1:T}^t, \boldsymbol{z}^t | \boldsymbol{x}^t, \boldsymbol{y}_0^t\right)} \left[ \log \frac{p(\boldsymbol{z}^t) p_\psi(\boldsymbol{x}^t \mid \boldsymbol{z}^t) p_\theta\left(\boldsymbol{y}_{0:T}^t \mid \boldsymbol{x}^t, \boldsymbol{z}^t\right)}{q_\rho(\boldsymbol{z}^t \mid \boldsymbol{x}^t, \boldsymbol{y}_0^t) q(\boldsymbol{y}_{1:T}^t \mid \boldsymbol{y}_0, \boldsymbol{z}^t, \boldsymbol{x}^t)} \right] \\
&= \mathbb{E}_{q\left(\boldsymbol{y}_{1:T}^t, \boldsymbol{z}^t | \boldsymbol{x}^t, \boldsymbol{y}_0^t\right)} \left[ \log \frac{p(\boldsymbol{z}^t)}{q_\rho(\boldsymbol{z}^t \mid \boldsymbol{y}_0, \boldsymbol{x})} + \log p_\psi(\boldsymbol{x}^t \mid \boldsymbol{z}^t) + \log \frac{p_\theta\left(\boldsymbol{y}_{0:T}^t \mid \boldsymbol{x}^t, \boldsymbol{z}^t\right)}{q(\boldsymbol{y}_{1:T}^t \mid \boldsymbol{y}_0^t, \boldsymbol{z}^t, \boldsymbol{x}^t)} \right] \\
&= \mathbb{E}_{q_\rho\left(\boldsymbol{z}^t | \boldsymbol{x}^t, \boldsymbol{y}_0^t\right)} \left[ \log \frac{p(\boldsymbol{z}^t)}{q_\rho(\boldsymbol{z}^t \mid \boldsymbol{y}_0^t, \boldsymbol{x}^t)} + \log p_\psi(\boldsymbol{x}^t \mid \boldsymbol{z}^t) \right] + \\
&\quad \mathbb{E}_{q\left(\boldsymbol{y}_{1:T}^t, \boldsymbol{z}^t | \boldsymbol{x}^t, \boldsymbol{y}_0^t\right)} \left[ \log \frac{p_\theta\left(\boldsymbol{y}_{0:T}^t \mid \boldsymbol{x}^t, \boldsymbol{z}^t\right)}{q(\boldsymbol{y}_{1:T}^t \mid \boldsymbol{y}_0^t, \boldsymbol{z}^t, \boldsymbol{x}^t)} \right] \\
&= \mathbb{E}_{q_\rho\left(\boldsymbol{z}^t | \boldsymbol{x}^t, \boldsymbol{y}_0^t\right)} \left[ \log p_\psi(\boldsymbol{x}^t \mid \boldsymbol{z}^t) \right] - D_{KL}\left(q_\rho\left(\boldsymbol{z}^t \mid \boldsymbol{x}^t, \boldsymbol{y}_0^t\right) \| p(\boldsymbol{z}^t)\right) + \\
&\quad \mathbb{E}_{\boldsymbol{z}^t \sim q_{rho}\left(\boldsymbol{z}^t | \boldsymbol{x}^t, \boldsymbol{y}_0^t\right)} \mathbb{E}_{q\left(\boldsymbol{y}_{1:T}^t | \boldsymbol{z}^t, \boldsymbol{y}_0^t\right)} \left[ \frac{p_\theta\left(\boldsymbol{y}_{0:T}^t \mid \boldsymbol{z}^t\right)}{q\left(\boldsymbol{y}_{1:T}^t \mid \boldsymbol{y}_0^t, \boldsymbol{z}^t\right)} \right] := \mathcal{L}_{ELBO}^l.
\end{aligned}
\tag{22}
$$

Analogously, we additionally impose the classifier $p_\omega(\boldsymbol{y}_0^t \mid \boldsymbol{z}^t)$ in the latent space to jointly train the source and target labeled data. The final training objective $\mathcal{L}_t^l$ for labeled target data is therefore:

$$\mathcal{L}_t^l = -\mathcal{L}_{ELBO}^l + \mathbb{E}_{(\boldsymbol{x}^t, \boldsymbol{y}_0^t) \sim \mathcal{T}_l} \mathbb{E}_{q_\rho(\boldsymbol{z}^t | \boldsymbol{x}^t)} [-\log p_\omega(\boldsymbol{y}_0^t \mid \boldsymbol{z}^t)]. \tag{23}$$

## A.3 Derivation of the Unlabeled Target Objective

For the unlabeled target data $\boldsymbol{x}^t$, our goal is to inference the low-dimensional latent embedding $\boldsymbol{z}^t$ that induces a domain-invariant latent space $\mathcal{Z}$ and the class label $\boldsymbol{y}_0^t$ based on $\boldsymbol{z}^t$. Besides, we assume a meanfield distribution on $q(\boldsymbol{z}^t, \boldsymbol{y}_0^t \mid \boldsymbol{x}^t)$, which can then be factorized as:

$$q(\boldsymbol{z}^t, \boldsymbol{y}_0^t \mid \boldsymbol{x}^t) = q(\boldsymbol{z}^t \mid \boldsymbol{x}^t) q(\boldsymbol{y}_0^t \mid \boldsymbol{x}^t). \tag{24}$$

Therefore, we optimize the following ELBO by regarding $\boldsymbol{z}^t$ and $\boldsymbol{y}_0^t$ as unknowns:

$$
\begin{aligned}
\log p\left(\boldsymbol{x}^t\right) &= \log \int p\left(\boldsymbol{x}^t, \boldsymbol{y}_0^t, \boldsymbol{z}^t\right) d\boldsymbol{y}_0^t d\boldsymbol{z}^t \\
&\geq \mathbb{E}_{q\left(\boldsymbol{y}_0^t, \boldsymbol{z}^t \mid \boldsymbol{x}^t\right)}\left[\log \frac{p\left(\boldsymbol{x}^t, \boldsymbol{y}_0^t, \boldsymbol{z}^t\right)}{q\left(\boldsymbol{y}_0^t, \boldsymbol{z}^t \mid \boldsymbol{x}^t\right)}\right] \\
&= \mathbb{E}_{q\left(\boldsymbol{y}_0^t, \boldsymbol{z}^t \mid \boldsymbol{x}^t\right)}\left[\log \frac{p(\boldsymbol{z}^t) p_\psi(\boldsymbol{x}^t \mid \boldsymbol{z}^t) p_\omega(\boldsymbol{y}_0^t \mid \boldsymbol{x}^t, \boldsymbol{z}^t)}{q_\rho\left(\boldsymbol{z}^t \mid \boldsymbol{x}^t\right) q\left(\boldsymbol{y}_0^t \mid \boldsymbol{x}^t\right)}\right] \\
&= \mathbb{E}_{q\left(\boldsymbol{y}_0^t, \boldsymbol{z}^t \mid \boldsymbol{x}^t\right)}\left[\log \frac{p(\boldsymbol{z}^t)}{q_\rho(\boldsymbol{z}^t \mid \boldsymbol{x}^t)} + \log p_\psi(\boldsymbol{x}^t \mid \boldsymbol{z}^t) + \log \frac{p_\omega(\boldsymbol{y}_0^t \mid \boldsymbol{x}^t, \boldsymbol{z}^t)}{q(\boldsymbol{y}_0^t \mid \boldsymbol{x}^t)}\right] \quad (25) \\
&= \mathbb{E}_{q_\rho(\boldsymbol{z}^t \mid \boldsymbol{x}^t)}[\log p_\psi(\boldsymbol{x}^t \mid \boldsymbol{z}^t)] - D_{KL}\left(q_\rho(\boldsymbol{z}^t \mid \boldsymbol{x}^t) \| p(\boldsymbol{z}^t)\right) + \\
&\quad \mathbb{E}_{\boldsymbol{z}^t \sim q_\rho(\boldsymbol{z}^t \mid \boldsymbol{x}^t)} \mathbb{E}_{q\left(\boldsymbol{y}_0^t \mid \boldsymbol{x}^t\right)}\left[\log \frac{p_\omega(\boldsymbol{y}_0^t \mid \boldsymbol{x}^t, \boldsymbol{z}^t)}{q(\boldsymbol{y}_0^t \mid \boldsymbol{x}^t)}\right] \\
&= \mathbb{E}_{q_\rho(\boldsymbol{z}^t \mid \boldsymbol{x}^t)}[\log p_\psi(\boldsymbol{x}^t \mid \boldsymbol{z}^t)] - D_{KL}\left(q_\rho(\boldsymbol{z}^t \mid \boldsymbol{x}^t) \| p(\boldsymbol{z}^t)\right) - \\
&\quad \mathbb{E}_{\boldsymbol{z}^t \sim q(\boldsymbol{z}^t \mid \boldsymbol{x}^t)} \underbrace{\left[D_{KL}(q(\boldsymbol{y}_0^t \mid \boldsymbol{x}^t) \| p_\omega(\boldsymbol{y}_0^t \mid \boldsymbol{x}^t, \boldsymbol{z}^t))\right]}_{②} := \mathcal{L}_{ELBO}^u.
\end{aligned}
$$

Since our deterministic classifier encodes the covariate-dependence between $\boldsymbol{y}_0^t$ and $\boldsymbol{z}^t$, therefore, $\boldsymbol{y}_0^t$ is not depended on $\boldsymbol{x}_t$ in our formulation, i.e., $p_\omega(\boldsymbol{y}_0^t \mid \boldsymbol{x}^t, \boldsymbol{z}^t) = p_\omega(\boldsymbol{y}_0^t \mid \boldsymbol{z}^t)$. ② demonstrates that, to maximize $\mathcal{L}_{ELBO}^u$, $D_{KL}(q(\boldsymbol{y}_0^t \mid \boldsymbol{x}^t) \| p_\omega(\boldsymbol{y}_0^t \mid \boldsymbol{z}^t) \equiv 0$ should always be satisfied. On the other hand, we empirically find that $p_\omega(\boldsymbol{y}_0^t \mid \boldsymbol{z}^t)$ can be a good approximation of $q(\boldsymbol{y}_0^t \mid \boldsymbol{x}^t)$ even when it is solely based on latent $\boldsymbol{z}^t$. Therefore, we assume that the model output $\boldsymbol{y}_0^t$ is only depended on the latent embedding $\boldsymbol{z}^t$, which supports the derivation in Eq. (20).

## B Algorithm

### B.1 Algorithm of DAPM-TT for Conventional Active Domain Adaptation

The overall trianing and section procedure of DAPM-TT for ADA is summarized in Algorithm 1.

---

**Algorithm 1** Pseudo code of DAPM-TT for ADA

---

**Require:** Labeled source dataset $\mathcal{S}$, whole target dataset $\mathcal{T}$, unlabeled target dataset $\mathcal{T}_u$, labeled target dataset $\mathcal{T}_l$, total training rounds $R$, total annotation budget $B$, per round annotation budget $b$, step number per adaptation stage $N_a$, step number per diffusion stage $N_d$.
**Ensure:** Optimal model parameters $\{\theta, \rho, \phi, \psi, \omega, \tau\}$.
1: Initialize student model parameters $\{\rho, \omega\}$ and other parameters $\{\theta, \phi, \psi, \tau\}, \mathcal{T}^l = \varnothing, \mathcal{T}^u = \mathcal{T}$
2: Initialize teacher model parameters $\Omega' = \{\rho, \omega\}$
3: **for** $t = 1$ **to** $R$ **do**
4:     **for** $i = 1$ **to** $N_a$ **do**
5:         Update parameters $\{\rho, \phi, \psi, \omega, \tau\}$ via optimizing Eq. (14).     % Adaptation Stage
6:         Update teacher model parameters $\Omega'$ with updated $\{\rho, \omega\}$ based on EMA.
7:     **end for**
8:     **for** $j = 1$ **to** $N_d$ **do**
9:         Update diffusion classifier parameters $\theta$ via optimizing Eq. (15).     % Diffusion Stage
10:     **end for**
11:     **if** $t \leq \frac{B}{b}$ **then**
12:         For each $\boldsymbol{x}^t \in \mathcal{T}_u$, generate $N$ predictions $\{\widetilde{\boldsymbol{y}}_n^t\}_{n=1}^N$     % Selection Stage
13:         Identify the two most predicted classes $a, b$ for each $\boldsymbol{x}^t$.
14:         Conduct t-test between $\{\widetilde{\boldsymbol{y}}_n^t[a]\}_{n=1}^N$ and $\{\widetilde{\boldsymbol{y}}_n^t[b]\}_{n=1}^N$ and obtain the p-value for each $\boldsymbol{x}^t$.
15:         $Selected \leftarrow$ Select samples with top-$b$ p-values from $\mathcal{T}_u$.
16:         $\mathcal{T}_u = \mathcal{T}_u \backslash Selected, \mathcal{T}_l = \mathcal{T}_l \cup Selected$.
17:     **end if**
18: **end for**
19: **return** Final model parameters $\{\theta, \rho, \phi, \psi, \omega, \tau\}$.

---

## B.2 Algorithm of DAPM-TT for Source-Free Active Domain Adaptation

In SFADA, We do not use the domain classifier since the source domain data and target domain data cannot co-exist. In addition, we also use confident unlabeled target samples that are pseudo-labeled by the teacher model to substitute the source-labeled samples in $\mathcal{L}_s$ in the diffusion stage. We denote the corresponding training loss by $\mathcal{L}_t^p$.

The overall trianing and section procedure of DAPM-TT for SFADA is summarized in Algorithm 2.

---

**Algorithm 2** Pseudo code of DAPM-TT for SFADA

---

**Require:** Labeled source dataset $\mathcal{S}$ for pre-training, whole target dataset $\mathcal{T}$, unlabeled target dataset $\mathcal{T}_u$, labeled target dataset $\mathcal{T}_l$, total training rounds $R$, total annotation budget $B$, per round annotation budget $b$, step number per adaptation stage $N_a$, step number per diffusion stage $N_d$, step number of source pre-training $N_s$.

**Ensure:** Optimal model parameters $\{\theta, \rho, \phi, \psi, \omega\}$.

1: Initialize model parameters $\{\rho, \omega\}$ and other parameters $\{\theta, \phi, \psi\}$, $\mathcal{T}^l = \varnothing$, $\mathcal{T}^u = \mathcal{T}$
2: **for** $i = 1$ **to** $N_s$ **do**
3:      Update source model parameters $\rho, \phi, \omega$ via optimizing $\mathcal{L}_s$.      % Source Pre-training
4: **end for**
5: Initialize teacher model parameters $\Omega' = \{\rho, \omega\}$
6: **for** $t = 1$ **to** $R$ **do**
7:      **for** $j = 1$ **to** $N_a$ **do**
8:          Update parameters $\{\rho, \psi, \omega\}$ via optimizing $\mathcal{L}_t^u + \mathcal{L}_t^l$.      % Adaptation Stage
9:          Update teacher model parameters $\Omega'$ with updated $\{\rho, \omega\}$ based on EMA.
10:      **end for**
11:      **for** $k = 1$ **to** $N_d$ **do**
12:          Update diffusion classifier parameters $\theta$ via optimizing $\mathcal{L}_t^p + \mathcal{L}_t^l$.      % Diffusion Stage
13:      **end for**
14:      **if** $t \leq \frac{B}{b}$ **then**
15:          For each $\boldsymbol{x}^t \in \mathcal{T}_u$, generate $N$ predictions $\{\widetilde{\boldsymbol{y}}_n^t\}_{n=1}^N$      % Selection Stage
16:          Identify the two most predicted classes $a, b$ for each $\boldsymbol{x}^t$.
17:          Conduct t-test between $\{\widetilde{\boldsymbol{y}}_n^t[a]\}_{n=1}^N$ and $\{\widetilde{\boldsymbol{y}}_n^t[b]\}_{n=1}^N$ and obtain the p-value for each $\boldsymbol{x}^t$.
18:          $Selected \leftarrow$ Select samples with top-$b$ p-values from $\mathcal{T}_u$.
19:          $\mathcal{T}_u = \mathcal{T}_u \backslash Selected$, $\mathcal{T}_l = \mathcal{T}_l \cup Selected$.
20:      **end if**
21: **end for**
22: **return** Final model parameters $\{\theta, \rho, \phi, \psi, \omega\}$.

---

# C More Implementation Details

## C.1 Network Architecture

**Variational Autoencoder** The architecture of the VAE and the deterministic classifier is presented in detail in Fig. 5. The encoder comprises a pre-trained ResNet-50 backbone and three initialized linear layers with a ReLU activation following the first linear layer. We assume a Gaussian distribution for the latent embedding, and its mean and covariance are estimated by two separate linear layers based on the first linear layer. The decoder is a two-layer MLP that has the same output dimension as the backbone's output. This encourages the decoder to reconstruct the feature generated by the backbone.

**Deterministic Classifier** The deterministic is simply a single layer linear classifier. We adopt the weight normalization technique on the classifier to stablize the training.

**Diffusion Classifier** The diffusion classifier is conditioned on the latent embedding $\boldsymbol{z}$, the ground-truth $\boldsymbol{y}$, the guided information $f_\Omega$ and the time step $t$. We adopt the same architecture as [19] for class variable diffusion, except for the dimension of the input variable. For clarity, we describe the detailed model structure in Fig. 6 (a).

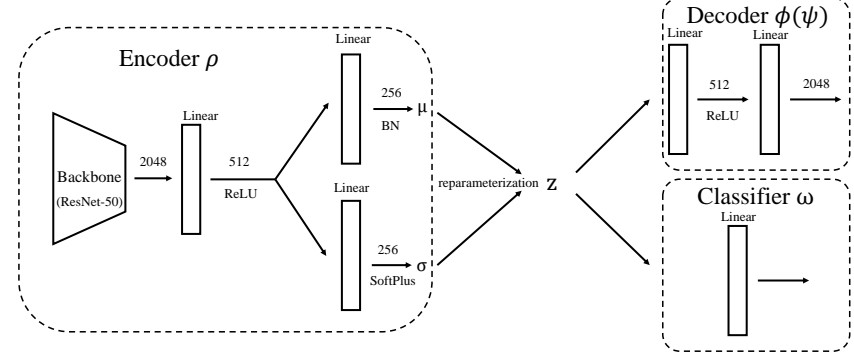

Figure 5: Architecture of the variational autoencoder and the deterministic classifier used in this work. The numbers on the data flow indicate the dimensions of the model output.

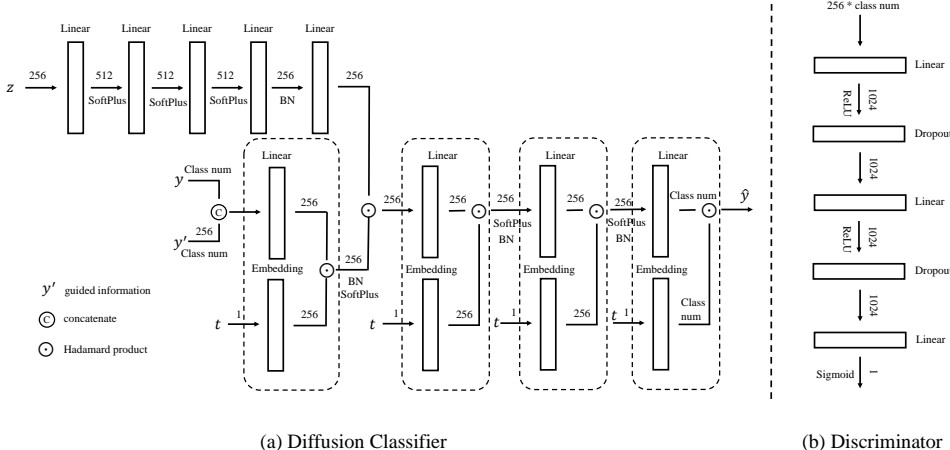

(a) Diffusion Classifier          (b) Discriminator

Figure 6: Architecture of (a) the diffusion classifier and (b) the domain discriminator.

**Domain Discriminator.** As shown in Fig. 6 (b). The domain discriminator we used is a three-layer MLP with a Dropout layer after the first and the second layers. And the output is a one-dimensional value with Sigmoid activation, which indicates the domainness of the sample.

## C.2 Training Details

**Conventional Active Domain Adaptation.** In the adaptation stage, we utilize the SGD optimizer with a learning rate of 0.01, momentum of 0.9, and weight decay of 0.001. We set the EMA rate for the teacher model to 0.99. In the VAE objective, we assume a standard Gaussian distribution, $\mathcal{N}(\mathbf{0}, I)$, for the prior distribution of the latent variable $z^s(z^t)$. For Office-31 and VisDA, we use the features generated by a pre-trained ResNet-50 and freeze the backbone parameters to accelerate training and conserve memory. However, for Office-Home, which has more diverse categories, we jointly train the backbone with other modules to learn more specific category knowledge, and we set the learning rate to 0.001, which is 10 times lower than that of other models. For Office-31 and Office-Home, we conduct adaptation for 5 epochs and train the diffusion classifier for 10 epochs in each training round. The total number of training rounds is 20. For VisDA, we set the epoch number in each stage to 1, and the total number of training rounds is 10. To train the diffusion classifier, we use the Adam optimizer with a learning rate of 0.001 and epsilon of 1e-8. The batch size is the same as that in the adaptation stage, and we use an EMA strategy with a rate of 0.9999 to update the model parameters. All experiments are conducted on a single RTX 3090 GPU.

**Source-Free Active Domain Adaptation.** In SFADA, we use the SGD optimizer without momentum and weight decay for adaptation. The learning rate is set to 0.01 for Office-31 and Office-Home, and 0.001 for VisDA. As with ADA, we freeze the backbone for Office-31 and VisDA and open it for

Office-Home. We pre-train the source model for 10 epochs for VisDA and 30 epochs for the other datasets. In each active learning round, we set the epoch numbers for the adaptation stage and the diffusion stage to 5 and 10, respectively, for Office-31 and Office-Home, and both to one for VisDA. The training details for the VAE, teacher model, and diffusion classifier are the same as in ADA.

### C.3 Hyperparameter Choices of the Diffusion Classifier

Following [19], the hyperparameters of the diffusion classifier are set in a standard DDPM [17] manner. Specifically, the number of diffusion timesteps $T$ is set to 1000, and a linear noise schedule with $\beta_1 = 0.0001$ and $\beta = 0.02$ is adopted for the forward diffusion process.

### C.4 Implementation of Compared Baseline Methods

Note that for conventional ADA, we cite the results of previous AL methods and ADA methods reproduced in [5] if the experimental settings are the same. For DUC [20] that does not report the result on Office-31 dataset, we report the resuls by our own runs based on the code from the official repository at `https://github.com/BIT-DA/DUC`. We have tuned some hyperparameters to ensure the best resuls we can achieve.

For SFADA, we implement compared baseline algorithms on our DAPM baseline with following details:

**Random.** We abondon the use of any selection strategy and randomly select samples from the unlabeled target dataset $\mathcal{T}_u$ for annotation.

**BvSB.** We compute the best-versus-second-best score based on the output of the deterministic classifier for each unlabeled sample and select $b$ samples with the lowest scores for annotation.

**Entropy.** We use the conditional entropy based on the output of the deterministic classifier to measure the prediction confidence. And samples with highest entropy values are selected for annotation.

**CoreSet.** We regard the sample selection in each round as a core-set cover problem and solve it with the code at `https://github.com/ozansener/active_learning_coreset`.

**BADGE.** We obtain the gradient vectors based on the pseudo labels generated by the deterministic classifier and utilize K-Means++ on the gradient vectors for diverse sampling. The algorithm is implemented based on the repository at `https://github.com/JordanAsh/badge`.

**ELPT.** We cite the resuls on Office-31 and Office-Home dataset from the original paper [47]. For VisDA, we run this method and report the resuls on ResNet-50 backbone based on the official code at `https://github.com/TL-UESTC/ELPT`.

## D   Additional Experimental Results

### D.1 Accuracies of Confident Predictions under Different Thresholds

Fig. 7 illustrates the accuracies of the teacher model's predictions for confident samples across different threshold settings. At each training step, the teacher model is updated with the student model, and we have computed the accuracy of the current batch and presented it as a curve. As expected, increasing the threshold leads to an increase in the teacher model's accuracy. However, when the threshold is relatively high, only a small number of samples are considered confident at the beginning, leading to a higher accuracy initially and a subsequent drop. It is worth noting that the teacher model provided relatively reliable predictions at a threshold value of 0.9. Raising the threshold further would result in too few confident samples, making 0.9 a more appropriate choice.

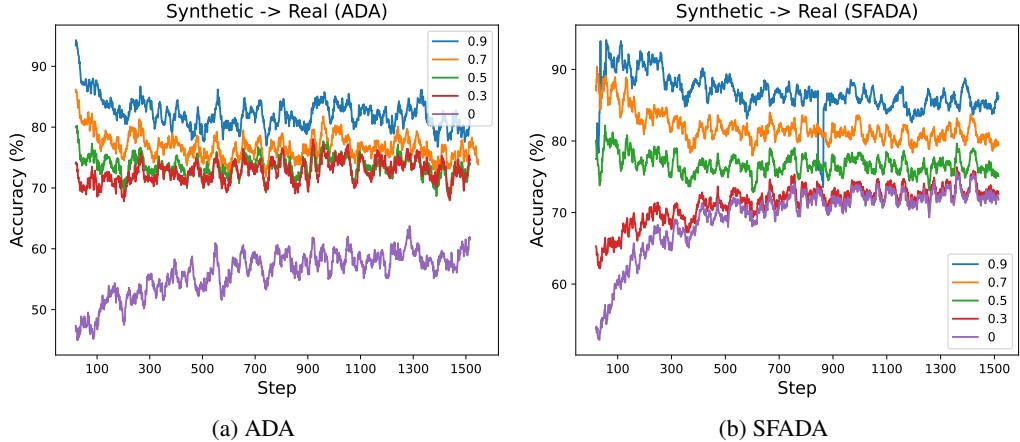

| (a) ADA | (b) SFADA |

Figure 7: Accuracies of confident predictions made by the teacher model under different threshold values on VisDA dataset.

## D.2 Prediction Visualization of Different Classifiers

To investigate the contrasting behaviors of deterministic and diffusive classifiers, we randomly select two samples on task Ar → Cl, and visualize the predictions made by the deterministic classifier and the diffusion classifier ($N = 100$). As shown in Fig. 8a, the deterministic classifier exhibits high confidence in predicting a $refrigerator$ as a $mug$. This verifies the overconfident issue in traditional softmax-based deterministic model, making it challenging for the active learning methods to detect the error and select such hard samples. In contrast, the diffusion classifier produces an uncertain prediction, indicating confusion in its output with a p-value of 0.873. As depicted in Fig. 8b, the deterministic classifier displays high uncertainty and misclassified the sample, whereas the diffusion classifier correctly classifies the sample with a p-value of 0.024, which saves budget and resources that would have been wasted on correcting the misclassification.

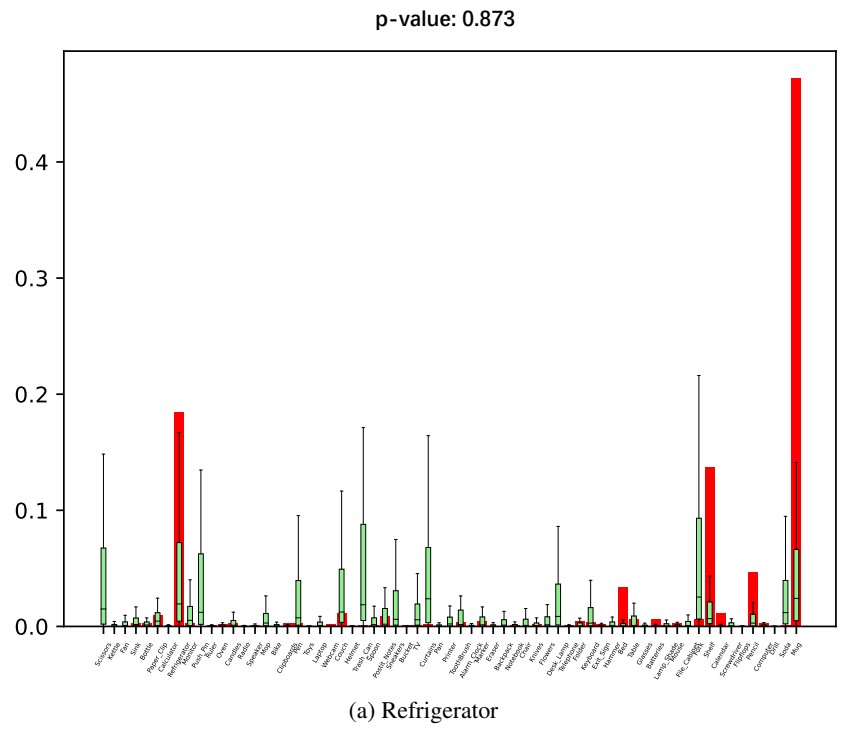

(a) Refrigerator

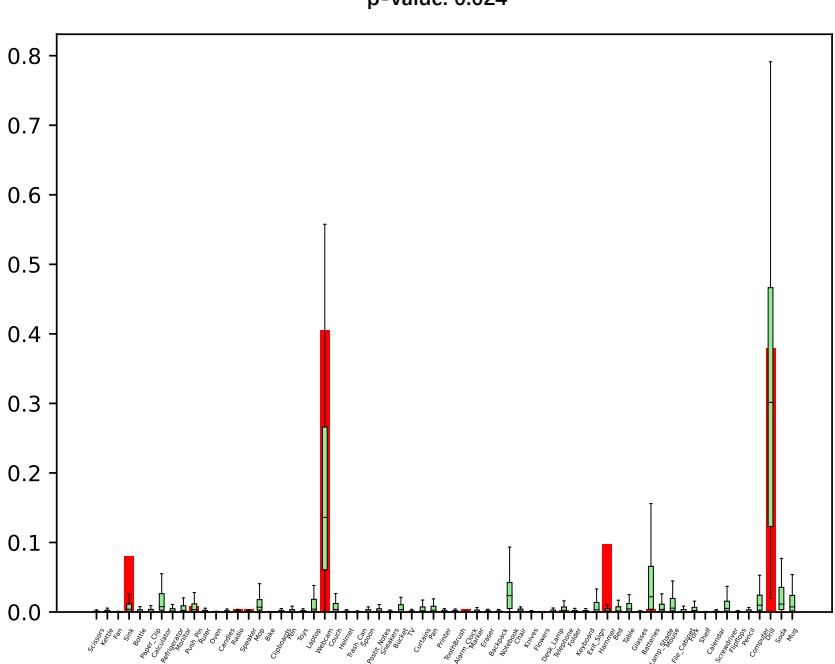

(b) Drill

Figure 8: Visualization of predictions made by different classifiers for 2 randomly picked samples from $refrigerator$ and $drill$, respectively. Red histograms represent the prediction of the deterministic classifier and green boxes denote the predictions of the diffusion classifier.

### D.3 Effect of the Modeling of the Latent Feature Distribution

In comparison to CARD [19] which utilizes a deterministic feature extraction network and a diffusion classifier to evaluate uncertainty based on the original image, our method employs an additional VAE to model data uncertainty in a low-dimensional latent space, and the diffusion classifier is based on latent variables. To investigate the benefits of this improvement for ADA tasks, we implement CARD in the ADA task and report the results of it on Office-31 and VisDA. Specifically, we use a deterministic ResNet-50 network as the feature extractor and train a deterministic classifier on top of it to guide the diffusion classifier. The diffusion classifier takes the original image $x$ as one of the inputs and uses the same independent two-sample t-test-based criterion for sample selection. We denote this implementation by CARD-TT. As shown in Table 5, DAPM-TT significantly outperforms CARD-TT on both datasets. It is exciting to see that although VAE is mainly designed for modeling the uncertainty of the data generation process, it results in a significant improvement with respect to accuracy. We conjecture that the reason for this improvement is two-fold: firstly, domain shift leads to a significant distribution shift in the image space, and in such case, CARD fails to work as intended [19]. This effect is mitigated to a certain extent in the low-dimensional and less noisy latent space. Secondly, in VAE training, we use the same prior distribution, i.e., the standard Gaussian distribution, for the latent variables of data in both the source and target domains. This design draws the samples in both domains closer to the standard Gaussian distribution, thereby achieving an indirect distribution alignment.

### D.4 Qualitative Analysis on Selected Samples

We present in Fig. 9 a list of all the selected samples by our approach to gain insight into which samples are chosen. Intuitively, our method tends to select samples that are challenging for the model, such as those with complex backgrounds or different styles from the other images in the dataset. Labeling these samples can help reduce the ambiguity in the model and enable it to better capture the semantic aspects of the category. Interestingly, we observe that our approach naturally selects a diverse range of sample classes, even though we did not explicitly impose a diversity constraint. Furthermore, we observe that the p-values of the selected samples gradually decrease with each

Table 5: Comparasion between probablistic and deterministic feature extractors, and between different t-test strategies on ADA task (ResNet-50). w/o and w/ are short for without and with, respectively.

| Category | Method | Office-Home | | | | | | | VisDA |
| | | A→D | A→W | D→A | D→W | W→A | W→D | Avg | Synthetic→ Real |
|---|---|---|---|---|---|---|---|---|---|
| w/o adaptaion stage | CARD-TT | 95.1 | 94.2 | 78.5 | 98.7 | 78.2 | 99.1 | 90.6 | 84.6 |
| | DAPM-TT | 96.1 | 95.9 | 79.5 | 98.7 | 79.2 | 99.1 | 91.4 | 86.3 |
| w/ adaptaion stage | DAPM-TT* | 96.8 | 97.8 | 83.3 | 99.8 | 81.7 | 100 | 93.2 | 88.6 |
| | DAPM-TT | 96.8 | 98.6 | 82.3 | 99.8 | 83.3 | 100 | 93.5 | 89.1 |

Table 6: Accuracy (%) of DAPM-TT combined with semi-supervised domain adaptation techniques on Office-Home with 5% annotation budget (ResNet-50).

| Method | Ar→Cl | Ar→Pr | Al→Rw | Cl→Ar | Cl→Pr | Cl→Rw | Pr→Ar | Pr→Cl | Pr→Rw | Rw→Ar | Rw→Cl | Rw→Pr | Avg |
|---|---|---|---|---|---|---|---|---|---|---|---|---|---|
| DAPM-TT (w/ CDAN) | 64.2 | 85.4 | 85.7 | 69.2 | 84.2 | 83.5 | 69.1 | 63.4 | 86.0 | 77.2 | 68.4 | 88.6 | 77.1 |
| DAPM-TT (w/ MME) | 65.1 | 85.5 | 84.9 | 72.5 | 84.6 | 83.7 | 72.4 | 64.7 | 86.3 | 77.9 | 70.3 | 88.8 | 78.1 |

training round, and by the 5th round, the lowest p-value is 0.518. This indicates that selecting approximately 5% of the samples is sufficient to mitigate much of the ambiguity in the model.

### D.5 Comparison between Different T-test Strategies

Based on the scores of the two most probable classes predicted by the diffusion classifier, we can use either paired two-sample t-test or independent two-sample t-test for selection, which correspond to different assumptions for the generation of predictions. The former assumes that the scores of different classes are generated in pairs, while the latter assumes that they are generated independently. For a paired t-test, the t-value of a target sample $x^t$ is calculated as follows:

$$t = (\bar{d} - \mu_d)/(s_d/\sqrt{N}), \tag{26}$$

where $\bar{d} = \frac{1}{N}\sum d_i$ is the mean of sample differences $d_i = \widetilde{y}_i^t[a] - \widetilde{y}_i^t[b]$, $\mu_d$ is the difference of the null hypothesis (usually set as 0), and $s_d = \sqrt{\sum(d_i - \bar{d})^2/N - 1}$ is the standard deviation of the sample difference.

We denote our method with paired t-test-based criterion by DAPM-TT*, and report the resuls on Office-31 and VisDA in Table 5. Empirically, we find that independent two-sample t-test yields superior performance on both datasets. We conjecture the reason might be that the independent two-sample t-test considers the internal variance of each group of samples. Therefore, when evaluating uncertainty, it considers an additional dimension compared to the paired t-test. In this work, we adopt independent two-sample t-test for all experiments.

### D.6 Combine with other Semi-Supervised UDA methods

It is worth noting that our formulation provides certain flexibility for the implementation of the training objectives, which showcases the modular nature of the framework. For instance, we use a teacher-student framework to implement the second term in Eq. (12) and adopt conditional adversarial learning as the main adaptation technique, which may limit the baseline DA performance and then affect the performance after active learning. For ADA, he DA component and AL component can be decoupled. We have tried to incorporate a SSDA method MME [62] to implement the adaptation stage, which is also practiced by previous ADA works, e.g., [11, 4]. Specifically, the adversarial learning loss (i.e., $\mathcal{L}_{adv}$) is replaced by the minimax entropy loss, and other losses remains unchanged. The results on Office-Home dataset can be found in Table 6, which shows the SSDA component further boosts the performance of our method.

### D.7 Benefits of Data-level Uncertainty

We conducted additional experiments ablating the VAE component, which can be simply implemented by replacing the variational encoder with a deterministic feature encoder. Through experiments, we found that the performance on VisDA-2017 is worse than modeling both the data-level and prediction-layer uncertainty, as shown in Table 7.

Table 7: Comparison between the performance of DAPM-TT w/ and w/o data-level uncertainty on VisDA-2017 dataset with 5% annotation budget (Resnet-50).

| Method | Synthetic $\rightarrow$ Real |
|---|---|
| DAPM-TT (w/ VAE) | 89.1 |
| DAPM-TT (w/o VAE) | 88.3 |

## E  Further Discussion

**Related Uncertainty Estimation Works.** In machine learning, there are primarily two kinds of uncertainties that are studied, i.e., *epistemic uncertainty* which arises due to a lack of knowledge or data and can be reduced with more data or improved models, and *aleatoric uncertainty* that stems from the inherent randomness in the data [51]. To model these uncertainties, the community has proposed many Bayesian deep learning methods. From a Bayesian perspective, these two uncertainties can be modeled by the posterior of model parameters $W$ and outputs $\boldsymbol{y}$, respectively, using the following formulation:

$$\mathrm{P}\left(\boldsymbol{y} \mid \boldsymbol{x}, \mathcal{D}\right) = \int \underbrace{\mathrm{P}\left(\boldsymbol{y} \mid \boldsymbol{x}, W\right)}_{\text{aleatoric uncertainty}} \underbrace{\mathrm{P}(W \mid \mathcal{D})}_{\text{epistemic uncertainty}} \, dW. \tag{27}$$

The family of Bayesian neural networks (BNNs) [48, 49, 63] is specifically designed to capture epistemic uncertainty by assuming a probability distribution over the network parameters. This involves estimating the posterior distribution over the parameters of the neural network given the observed data. However, due to the intractable form of the posterior, BNNs are often trained using appropriate approximations like Markov Chain Monte Carlo (MCMC) or Variational Inference (VI). Another approximation for BNNs is Monte Carlo Dropout [64], which assumes a Bernoulli distribution over network parameters. During inference, the output of the network is averaged over multiple stochastic forward passes with dropout enabled, resulting in a distribution over the predictions.

Evidential deep learning (EDL) [50, 20] is another method that models uncertainty associated with the output of the model based on evidential theory. In EDL, this is typically achieved by using a distributional output based on the theory of Subjective Logic [65], such as a Dirichlet distribution over class probabilities for classification tasks, instead of a point estimate.

For non-Bayesian methods, ensemble-based methods [66, 67] have been proposed to model predictive uncertainty by combining multiple deterministic neural networks with different initializations. However, all these methods are designed to capture either epistemic uncertainty or aleatoric uncertainty alone by modeling probability distributions over the model parameters and outputs, respectively. Moreover, they still impose a restricted form of distributions, such as Gaussian or Dirichlet, which limits their applicability in practice.

To capture both sources of uncertainties in a single model, Kendall et al. [51] propose modeling aleatoric uncertainty in the model outputs beyond model parameters by predicting the noise term for the output variable of each sample as part of the model output. However, the form of the noise is still assumed to be Gaussian.

Recently, Han [19] proposed modeling the implicit output distribution by leveraging the generative capability of the diffusion model. However, they only model aleatoric uncertainty in their formulation since the model they use is still a deterministic neural network, and the proposed method, CARD, only enables in-distribution generalization.

In addition to modeling aleatoric uncertainty with the diffusion classifier, we also incorporate a VAE to model the underlying data generation process. The VAE learns a probabilistic distribution over the latent space, which represents the model's uncertainty about the true underlying distribution of the data, given the limited amount of training data. As more data is provided during training, the learned distribution should converge to the true underlying distribution, reducing epistemic uncertainty. Therefore, our DAPM also offers a way to measure epistemic uncertainty. Furthermore, our diffusion classifier is conditioned on the latent variables in $\mathcal{Z}$ rather than the ones in the original image space $\mathcal{X}$. We argue that the latent space contains less noise and is more suitable for cross-domain tasks.

**The choice of Diffusion-based model for probabilistic modeling.** Diffusion models have been shown to be equivalent to reversing stochastic differential equations (SDEs). Running the reverse process along the SDE trajectory to recover the original data distribution provides a principled way to recover the data distribution. Besides, compared to other generative models that complete generation in a single step, the multi-step noise schedule smooths the posterior and improves the exploration of the probability space, which will also result in more stable training and generation. In addition, the forward-reverse diffusion process allows effective modeling of high-dimensional complex distributions without parametric assumptions. Actually, the correlation between the covariates and the prediction variable is assumed to be high-dimensional and complex as well. Therefore, we regard diffusion models as a better choice to model the underlying predictive distribution.

# F   Limitations and Broader Impacts

**Limitations.** Our work presents a way to recover the predictive distribution of deep models by combining the power of diffusion models and variational autoencoders. However, like any research, our study may have some limitations that should be acknowledged. Firstly, the task we focus on is limited to image classification in this work. In our future study, we may extend the scope of research to other areas like image segmentation and object detection, etc. Secondly, probabilistic models are often more difficult to interpret compared to deterministic models. It is important to study insights in future research to help interpret probabilistic models.

**Broader Impacts.** Indeed, the impact of active domain adaptation is significant, especially in scenarios where labeled data is scarce in the target domain. The ability to adapt to new domains with limited labeled data can potentially reduce the time and cost required to gather labeled data for each specific task, thus making the deployment of machine learning models more accessible and cost-effective. This can also facilitate the development of more robust and generalizable machine learning models that can be used across multiple domains, which is particularly important for applications that operate in dynamic and diverse environments. Overall, our work contributes to advancing the field of machine learning and promoting the development of more efficient and adaptive technologies.

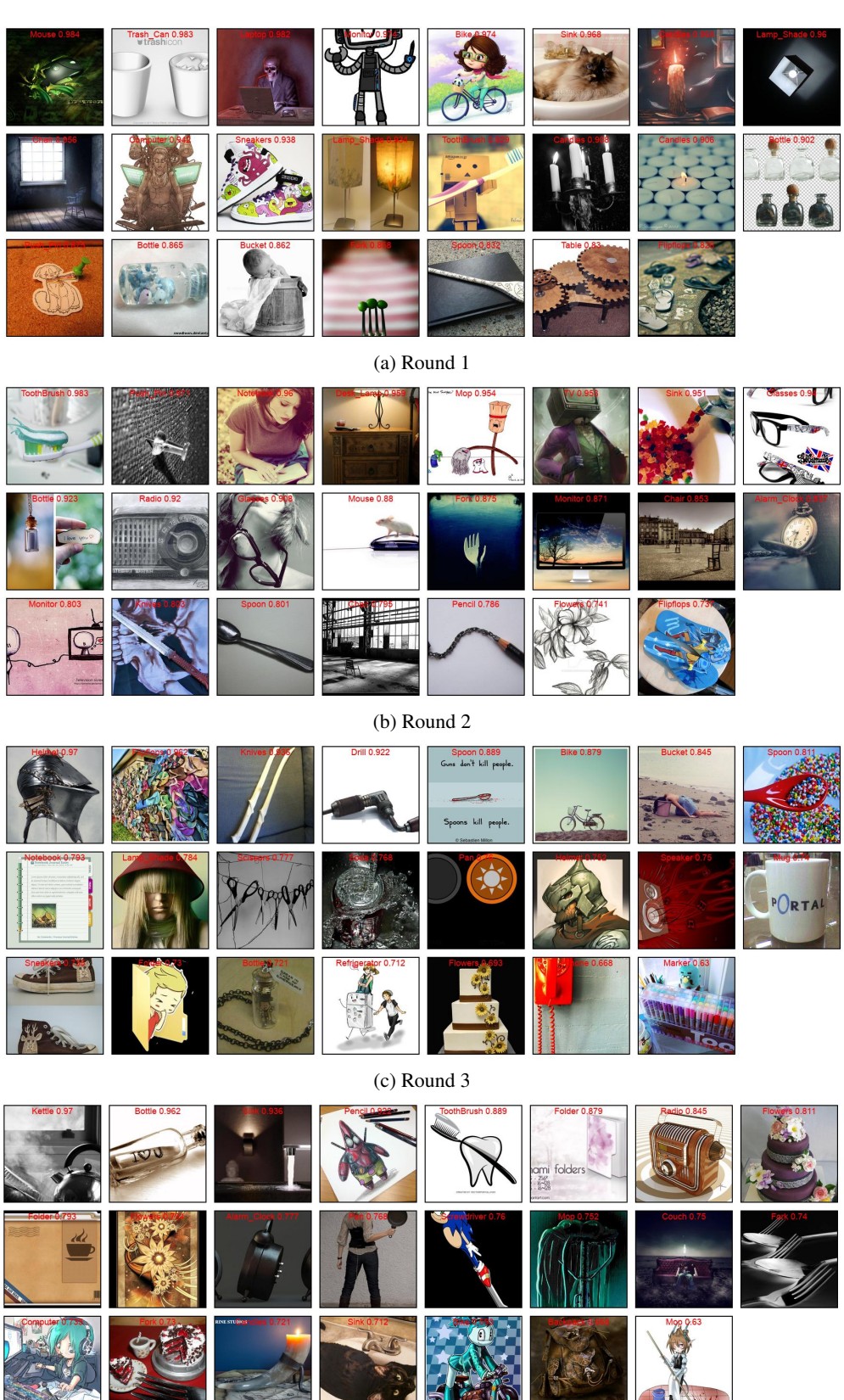

(a) Round 1

(b) Round 2

(c) Round 3

(d) Round 4

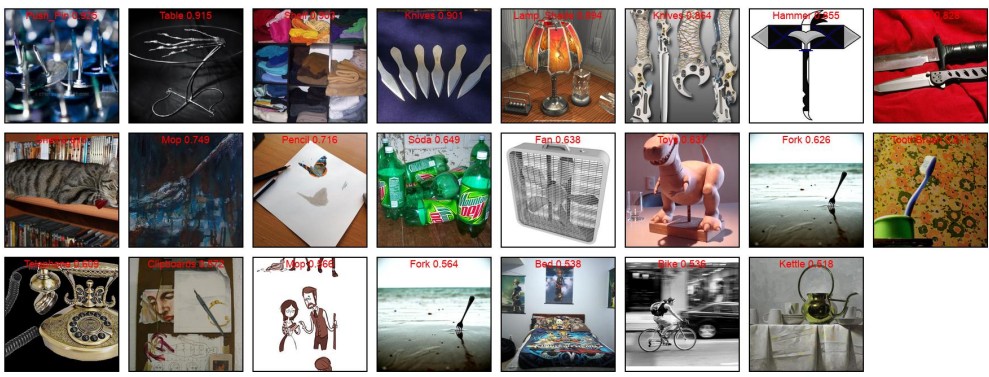

(a) Round 5

Figure 9: All selected samples on task Cl $\rightarrow$ Ar (ADA). For samples in each round, the priority (p-value) gradually decreases from top left to bottom right.

