# OpenReview forum: "Diffusion-Based Probabilistic Uncertainty Estimation for Active Domain Adaptation"
_NeurIPS.cc/2023/Conference — NeurIPS 2023 poster_

### Official Review · Reviewer_1fQe · 2023-06-14

**Soundness:** 3 good
**Presentation:** 3 good
**Contribution:** 2 fair
**Rating:** 5
**Confidence:** 4

**Summary:**

This paper tackles active domain adaptation based on the diffusion probabilistic model to estimate the prediction uncertainty. The authors apply the variational inference to the joint distributions of input data, latent codes in feature and prediction spaces, and labels. To realize adaptation, KL loss, and adversarial training are employed. Based on the losses, the model is trained in a two-stage fashion. To select the informative samples, a t-test-based strategy is explored.

**Strengths:**

* Estimating the prediction uncertainty based on diffusion models seems to be more accurate as shown in experiments.
* The results for active domain adaptation and source-free domain adaptation on three benchmarks are better than those of the baseline.
* The paper is well-written and easy to follow.


**Weaknesses:**

* The main weakness is the combination nature. The proposed method seems to be a combination of VAE, diffusion-based uncertainty estimation [19], knowledge distillation, and adversarial training, which are common techniques in generative models and domain adaptation. Such a contribution is below the standard of NeurIPS.
* The diffusion-based uncertainty estimation is borrowed from [19]. It would be better to develop it specified for DA with distribution shift.
* Missing results on more challenging datasets such as DomainNet. From Table 1-2, DAPM-TT is marginally better or worse than DUC on VisDA and Office-Home. It would be better to evaluate DAPM-TT and DUC on more challenging datasets.
* From Table 4, the results of the diffusion classifier and deterministic classifier are similar, so the effectiveness of the diffusion-based uncertainty estimation for ADA is limited.


**Questions:**

* The motivation for the design of the t-test-based strategy is not clear. Why did the authors choose the two most predicted classes? How about three or even more? Why define the t-value as Eq. (17)? Does it follow a t-distribution?
* With Eq. (17), how to obtain the p-value. Please provide more details.
* It would be better to add more explanation on the last two losses in Eq. (13).
* In Line 40, the paper aims to accurately estimate the posterior distribution. However, a large margin exists between the estimated uncertainty and the true one, as in Fig. 2(a).


* Minor: How about the performance on modern backbones, such as transformer or CLIP?


**Limitations:**

It would be better to develop diffusion-based uncertainty estimation specified for DA with distribution shift.

---

> ### Author Rebuttal · Authors · 2023-08-08
>
> Dear Reviewer 1fQe:
>
> Thanks for your comments. Please kindly find below our responses and answers. Due to limited length, Table R6 and R7 are in the pdf file attached in the Common Response.
> ```
> Weakness 1: The combination nature.
> ```
> Thanks for the comment. We hope to clarify that the contribution of this work goes beyond a trivial combination of existing techniques. While we integrate several components, the main contribution lies in the unified probabilistic framework specifically designed for ADA using variational inference and diffusion models to capture both data and prediction uncertainties. The uniqueness resides in the thoughtful integration of these tailored components within a principled probabilistic framework, as noted by Reviewer 3siL in the Strength 1. Besides, the t-test-based selection is a novel strategy designed for ADA, resulting in clear empirical gains that show component synergy. We believe formulating ADA in a probabilistic manner is an important direction, and our framework also provides certain flexibility for implementation (Common Response 3). This provides a strong basis for further extensions accounting for other uncertainties or techniques.
> ```
> Weakness 2: It would be better to specifically develop methods for distribution shift.
> ```
> Thanks. In this work, our focus is exactly on developing diffusion-based uncertainty estimation under distribution shift. While the diffusion-based classification model draws inspiration from [19], we have innovatively tailored it for DA. First, the VAE aims to model the common generation process of both source and target data, which also captures data-level uncertainties in a latent space better suited for cross-domain tasks compared to the original input space used in [19]. Besides, the diffusion model is conditioned on this normalized latent space and enable effective domain transfer. In addition, the two-stage training first aligns domains before diffusion modeling for a shared embedding. We will highlight these key differences from prior work in the revised paper.
> ```
> Weakness 3 & Minor: Performance on more challenging datasets and modern backbones.
> ```
> Thanks. This is an interesting suggestion. Unfortunately, we do not yet have comprehensive results using other datasets and networks due to the significant computational requirements and time needed during the rebuttal to evaluate baselines and our method. However, we have begun some initial experiments on a more challenging dataset miniDomainNet, and on a Swin-Transformer backbone, which we can share some preliminary results in Table R6 and R7, respectively. We believe these initial results help demonstrate the availability of our method on more challenging datasets and modern networks.
> ```
> Weakness 4: The diffusion and deterministic classifiers perform similarly.
> ```
> We would like to clarify that there is a factual misinterpretation of the experimental result. The similarity between the deterministic and diffusion classifiers in Table 4 cannot imply the inefficacy of our uncertainty estimation method. Our method consists a deterministic and a diffusion classifier, either of which can be used for evaluation after training. Their accuracies on the target domain might be comparable, given that the diffusion classifier is generative and not trained with a discriminative objective. However, regardless of the evaluation classifier, diffusion-based uncertainty estimation is consistently used for sample selection during training.
> ```
> Q1&Q2: Motivation of the t-test-based strategy and details of the p-value.
> ```
> Focusing on the top 2 classes is a popular way to qualify prediction uncertainty. This approach, known as BvSB[36], is effective for uncertainty estimation in previous softmax-based deterministic models and AL. Considering more classes becomes less tractable for statistical analysis and insignificant. Our method employs the diffusion classifier to generate multiple stochastic predictions per instance. Therefore, rather than comparing the difference between two scalars, we need to compare differences between two groups of predictions. We think t-test is a good choice and have explained the merits of this design in Sec 3.4.
>
> Our t-value formulation aligns with the traditional two-sample t-test for comparing two independent groups. In our case the sample numbers N in two groups are equal, so the formulation comes to Eq. (17). With the calculated t-value, we treat it as coming from a t-distribution with degrees of freedom 2N–2. The CDF of the t-distribution yields the p-value, indicating the probability of observing a t-value at least as extreme as the one computed under the null hypothesis. Since this process is the same as traditional t-test, we omit the illustration. However, we would like to add more details.
> ```
> Q3: More explanation on the two losses in Eq. (13).
> ```
> The last loss encourages individual predictions on the unlabeled target samples to be confident, by maximizing the KL divergence between the predicted class distribution and a uniform distribution, which avoids high entropy outputs. The second last loss aims to make the predictions diverse across the whole unlabeled set, by minimizing the divergence between the dataset-level class distribution and a uniform distribution. Together, these losses allow us to produce target predictions that are individually confident but holistically diverse.
> ```
> Q4: Large gap in Fig. 2(a).
> ```
> While we strive for accurate posterior estimation, we recognize it is an extremely challenging task, particularly under domain shift. Our method makes progress but does not claim to perfectly solve this problem. We agree the curves in Fig. 2(a) indicates there is room for improvement. However, our method makes notable progress towards this goal. As shown in Fig. 2(a), the ECE (in the legend) is significantly smaller than the deterministic classifier, and also smaller than DUC. Such an ECE is a reasonably good calibration error in general.

---

> > ### Comment · Reviewer_1fQe · 2023-08-12
> > **Response to Authors**
> >
> > I appreciate the effort of the authors in the rebuttal. After reading the comments of the other reviewers and the response from the authors, I think the combination framework is somewhat non-trivial, and the t-test-based selection seems novel. Therefore, I decide to raise the score to borderline accept.

---

### Official Review · Reviewer_rDAp · 2023-06-29

**Soundness:** 3 good
**Presentation:** 3 good
**Contribution:** 3 good
**Rating:** 8
**Confidence:** 4

**Summary:**

This paper introduces a probabilistic framework for ADA, which incorporates both data-level and prediction-level uncertainties using variational inference. It employs a VAE and a diffusion-based classifier to approximate the joint posterior distribution. Adversarial learning is used to ensure an invariant latent space, and a t-test-based criterion is applied to select informative target samples. Experimental results demonstrate the effectiveness of the method on ADA and SFADA tasks.

**Strengths:**

1. This paper introduces a novel probabilistic framework for ADA, which goes beyond traditional deterministic model-based uncertainty estimation by capturing both data-level and prediction-level uncertainties. This approach is based on variational inference and incorporates a variational autoencoder and a latent diffusion classifier to approximate the joint posterior distribution.

2. This paper highlights that the proposed method provides more calibrated predictions compared to previous ADA methods. Calibrated predictions are desirable in ADA tasks to ensure reliable and well-calibrated confidence estimates for informativeness assessment.

3. The incorporation of t-test-based criterion for sample selection is inspiring and reasonable, as the t-value considers various factors, such as prediction variability, cross-category ambiguity, and sample size, in a unified manner, making it a comprehensive criterion.

4. The proposed method is naturally compatible with SFADA setting, since the diffusion-based classifier is built on top of a transferable task-specific model, which can be trained in a standard UDA manner or in a source-free UDA manner.


**Weaknesses:**

1. This paper presents a comprehensive framework that combines variational inference, adversarial learning, and knowledge distillation. While these techniques can enhance the capabilities of the model, they may introduce increased complexity and computational cost, which could limit the scalability and practicality of the method.

2. The introduced improvements of the proposed method over other methods appear to be limited. In particular, the proposed DAPM does not achieve the best performance on the Office-Home dataset, which necessitates further analysis or justification.

3. In the ablation study, the comparison primarily focuses on highlighting the improvements of the diffusion classifier over the deterministic classifier, specifically emphasizing the advantages of prediction-level probabilistic modeling. However, the benefits resulting from data-level probabilistic modeling brought by the VAE are not demonstrated in this analysis.


**Questions:**

1. I like the design of this work. However, it still remains unclear that why the diffusion-based classifier can accurately recover the predictive distribution. Is there any theoretical guarantee or intuitive justification that the diffusion-based classifier is a better choice to model the predictive distribution?

2. Does the use of the diffusion model and the two-stage training strategy significantly increase the training and inference time?

3. Due to the impracticability of conducting explicit adversarial learning within the SFADA setting, the term "DAPM" is unsuitable in this context, necessitating a more appropriate name specifically for SFADA.


**Limitations:**

The limitations have been discussed in the appendix. I do not find potential obvious negative social impact associated with this work.

---

> ### Author Rebuttal · Authors · 2023-08-08
>
> Dear Reviewer rDAp:
>
> Thanks for your positive words and constructive comments. Below are our responses to the weaknesses and questions.
> ```
> Weakness 1: Increased complexity and computational cost, limited scalability and practicality.
> ```
> Response: Thanks for your comments. The computational complexity is a widely concerned problem in the field of diffusion model-related research. To explore the complexity of our model, we have tested model size and inference speed. Please refer to the Common Response 2 for details.
>
> In addition, in light of Reviewer Grmw, we further investigate the modular nature and flexibility of our method, which show certain scalability and practicality of the probabilistic framework. Please refer to the Common Response 3 for details.
> ```
> Weakness 2: Limited improvements of the proposed method over other methods.
> ```
> Response: Thanks for your comments. As we illustrated in the Common Response 3, although the diffusion-based sample selection strategy is effective, since our framework mainly focuses on modeling uncertainty, therefore the DA baselines without active learning are not particularly impressive (see Tables R1 and R2), which is one of the main reasons why the final results are limited. Nevertheless, in order to demonstrate the effectiveness of our active learning approach in the DA scenario, we tried to combine a semi-supervised learning method (see Common Response 3), and the results do show that it can bring more performance improvements.
> ```
> Weakness 3: The benefits of data-level uncertainty is not clear.
> ```
> Response: Thanks for raising this point. According to your suggestion, we conducted additional experiments ablating the VAE component, which can be simply implemented by replacing the variational encoder with a deterministic feature encoder. Through preliminary experiments, we found that the performance on VisDA-2017 is worse than modeling both the data-level and prediction-layer uncertainty, as shown in Table R5. Due to extensive experimental requirements and limited time available during rebuttal, we are still working to try the performance on other datasets, which contains multiple adaptation tasks. We will attach it as a separate ablation analysis in the revised appendix.
>
> **Table R5** Comparison between the performance of DAPM-TT w/ and w/o data-level uncertainty on VisDA-2017 dataset (Resnet-50, 5% budget).
>
> | Method            | Synthetic → Real |
> |-------------------|------------------|
> | DAPM-TT (w/ VAE)  | 89.1             |
> | DAPM-TT (w/o VAE) | 88.3             |
> ```
> Q1: Why the diffusion-based model is a better choice?
> ```
> Response: Thank you for raising this excellent question. While it is difficult to have a full theoretical justification for this choice, we can justify diffusion models as a principled approach to implicitly recover predictive distributions in a flexible nonparametric manner through stochastic diffusion processes.
>
> Diffusion models have been shown to be equivalent to reversing stochastic differential equations (SDEs). Running the reverse process along the SDE trajectory to recover the original data distribution provides a principled way to recover the data distribution. Besides, compared to other generative models that complete generation in a single step, the multi-step noise schedule smooths the posterior and improves the exploration of the probability space, which will also result in more stable training and generation. In addition, the forward-reverse diffusion process allows effective modeling of high-dimensional complex distributions without parametric assumptions. Actually, the correlation between the covariates and the prediction variable is assumed to be high-dimensional and complex as well. Therefore, we regard diffusion models as a better choice to model the underlying predictive distribution.
> ```
> Q2: How does the diffusion model increase the training and inference time?
> ```
> Response: Thanks for your comments. We have analyzed the model size and the inference time of our method by comparing it with other baselines in Common Response 2. For the training time, training the diffusion model does not introduce a significant increase in time duration. Because for each, we sample a time step, the specific form of the forward diffusion process enables an efficient sampling for arbitrary steps in a closed form. The model is then used to predict the noise. In other words, we do not need to go through a full sampling trajectory for each sample, as in the inference phase, and repeat the process multiple times for generating predictions. In addition, as we illustrated in Appendix C, we do not train the ResNet-50 backbone in most experiments. Therefore, the additional training time involved by the diffusion model is not obvious.
> ```
> Q3: Inappropriate name of the method in SFADA.
> ```
> Response: Thanks for pointing out. We would like to rename our method as Diffusion-based Probabilistic Model (DPM) for SFADA in the revised paper.

---

> > ### Comment · Reviewer_rDAp · 2023-08-12
> > **My concerns have been effectively addressed, and I am inclined to increase my score.**
> >
> > After thoroughly reviewing the questions raised by other reviewers and carefully considering the authors' responses, I am convinced that my concerns have been effectively addressed. As a result, I am inclined to increase my score for this paper.

---

### Official Review · Reviewer_Grmw · 2023-07-04

**Soundness:** 3 good
**Presentation:** 3 good
**Contribution:** 3 good
**Rating:** 7
**Confidence:** 4

**Summary:**

This paper aims to tackle the uncertainty estimation problem in ADA with a probabilistic framework, which captures two levels of uncertainties by a variational autoencoder and a diffusion-based classifier, respectively. An adversarial adaptation strategy and a statistical test-based criterion are further introduced. The experiments are conducted on both ADA and source-free ADA tasks, which verify the effectiveness of the proposed DAPM.

**Strengths:**

1. This paper revolves around the fundamental question in ADA, i.e., uncertainty estimation. Specifically, the authors establish a probabilistic model to assess the full predictive distribution beyond a point estimate, which is accomplished by a conditional diffusion-based probabilistic model.

2. To tackle the domain shift problem, the authors propose to explore an invariant latent space with an adversarial learning strategy. The diffusion-based classifier is conditioned on latent variables, and the data uncertainty in this latent space is also explicitly modeled.

3. This paper is well organized and technically solid. Specifically, the motivation of this work is well established, and the proposed method is plausible to address the uncertainty estimation problem. Besides, the proposed method is compatible with both conventional ADA and source-free ADA settings as presented in the manuscript.

4. The experiments on three ADA datasets demonstrate that the proposed method exhibits not only high accuracy for classification but also low ECE for uncertainty calibration.

**Weaknesses:**

1. Despite the uncertainty estimation component (the second stage), the domain adaptation component (the first stage) is solely based on conditional domain adversarial networks (for ADA). This limited design could explain the marginal improvement observed when compared to other SOTA methods. I think the proposed method can benefit from incorporating other more advanced UDA or SSDA techniques for both ADA and SFADA.

2. The proposed method differs from previous adversarial learning-based ADA methods by employing adversarial learning between labeled and unlabeled samples. However, the underlying motivation for this design choice is not explicitly stated, and the potential benefits of it are not adequately evaluated or demonstrated.

3. Minor: The equation indexes in the appendix are missing.

**Questions:**

1. The diffusion model involves numerous steps of forward and inverse diffusion. How different are the model size and running time compared to baseline methods?

2. The diffusion-based classifier is conditioned on the output scores of the deterministic classifier, which acts as prior knowledge regarding the relationship between x and y_0. How would the model perform without this prior?

**Limitations:**

The authors have discussed the limitations and broader impacts in the appendix.

---

> ### Author Rebuttal · Authors · 2023-08-08
>
> Dear Reviewer Grmw,
>
> We appreciate the comments and questions you post. Here are our point-to-point responses.
> ```
> Weakness 1: Suboptimal domain adaptation component.
> ```
> Response: Thank you for the thoughtful feedback on improving the domain adaptation component of our approach. We totally agree that relying solely on adversarial alignment may limit the potential gains compared to state-of-the-art techniques. According to your suggestion, we have explored incorporating a more advanced SSDA method MME into the adaptation stage, and found this can provide further improvements. Please refer to the Common Response 3 for more details. These results align well with your insight about strengthening the adaptation component beyond adversarial learning. We will integrate a discussion of these experiments and analysis into the appendix to demonstrate the benefits of combining the uncertainty estimation approach with other domain adaptation techniques.
> ```
> Weakness 2: Why conduct adversarial learning between labeled and unlabeled data?
> ```
> Response: Thanks for the comment. The key motivation for aligning labeled and unlabeled data is to create a shared feature space that allows the model to effectively discriminate on labeled source/target data and produce meaningful uncertainty estimates on unlabeled target data. By aligning labeled and unlabeled, we get an invariant space for both supervised classification and unsupervised uncertainty modeling. In contrast, aligning domains directly may retain some divergence between labeled and unlabeled data distributions, since the labeled target samples are incorporated into the supervised training objective with labeled source samples. We will explain the motivation more clearly in the revised paper.
> ```
> Minor: Missing equation indexes.
> ```
> Response: Thanks for recognizing this. We have fixed this issue in the revised appendix.
> ```
> Q1: Model size and running time compared to baseline methods.
> ```
> Response: Thanks for the question. Computational complexity is a problem that is widely concerned in diffusion models. We have re-run our method to check the running time and model size of our method as well as other baseline methods. Please refer to the Common Response 2 for more details.
> ```
> Q2: How does the model perform without conditioning on the output scores of the deterministic classifier?
> ```
> Response: Thank you for raising this insightful question. In our early work, we did conduct experiments to evaluate the effect of this design choice, where the diffusion model is not conditioned on the deterministic classifier outputs, instead, a standard Gaussian distribution is used. However, we empirically found removing this prior knowledge resulted in 0.8% drop in mean accuracy on the Office-31 dataset, and 1.1% drop on the VisDA dataset. We conjecture the prior helps guide the diffusion model to produce higher quality uncertainty estimates and accelerate learning. This observation also aligns with the finding in [19], where the diffusion model is employed in an in-distribution scenario. Therefore, we also incorporate the prior knowledge as the condition of the diffusion model in this work.

---

### Official Review · Reviewer_fVMQ · 2023-07-06

**Soundness:** 3 good
**Presentation:** 3 good
**Contribution:** 3 good
**Rating:** 6
**Confidence:** 3

**Summary:**

Authors propose a new model for active domain adaptation. They train a VAE with domain-invariant latents (using adversarial alignment) and a supervised+teacher/student classifier head on top of learned embeddings to obtain point estimates of class probabilities, and then train a denoising diffusion model on predicted class labels to more accurately predict uncertainty of predictions. More specifically, for each unlabeled target sample, authors use denoising diffusion to sample from the distribution of predicted softmax vectors, then compute t-test statistic for the null hypothesis that two most likely output classes have equal probabilities, and label examples that fail to reject this hypothesis the most. Authors evaluate this (and an analogous source-free algorithm) on VisDA and Office-Home and show that in most cases their method achieves best performance (except Active DA on Office-Home where it archives 2nd best) and is calibrated better then the runner up (DUC). Authors evaluate the effect of the number of diffusion samples, the annotation budget, they propose and check several alternative deterministic and diffusion-based selection criteria, hyperparameter sensitivity, and effect of different VAE losses (e.g. adversarial alignment).

**Strengths:**

The paper is well-written, technically sound, does a good job explaining the method, and the performance is good. I had several questions and ablation in mind while reading the paper, and most of them were addressed in the experiment section. The evaluation is thorough.



**Weaknesses:**

The method uses every trick from the book to beat state of the art (eg student/teacher training, etc.), and combined with the proposed novel approach (diffusion-based selection criterion), it indeed achieves a small improvement over sota. I appreciate that authors provide this detailed ablation, but it reveals that the novel component provides only ~1pp improvement over a very strong “deterministic backbone method”.

I understand that authors attempted to squeeze as much content as possible into the paper, but currently the paper is at times difficult to read because the table / figure margins are so thin.

In several places authors use a(b) to mean "a (or b)" which is usually clear from context but might be confusing for some readers.

Why in Figure 1 both green (labeled source) and pink (all labeled) go to \tau? Isn't green included in pink?

Also, see questions below.



**Questions:**

1. Diffusion sampling might be time-consuming. How does the runtime/training time compares to other methods? Or at least the deterministic backbone? (+10%? 2x? 10x?)

2. Why is DUC not shown in Figure 3a?

3. I wanted to ask, given how thin margins are in Table 3, and given that “your” best deterministic strategy seems to beat most of SotA deterministic baselines, whether improvement should be attributed mostly to the proposed “diffusion uncertainty estimation” or a more powerful backbone. But then I realized that I cannot find number 93.5 from Table 3 in other tables. Where can I find corresponding baseline performances?

4. On ADA method beats competing methods in low-data regime, but on SFADA it does not - any thoughts on why this is the case?

5. It might be possible to obtain nonparametric estimates of entropy from sampled predictions.



**Limitations:**

Authors did not explicitly address limitations and societal impact in the main paper.

---

> ### Author Rebuttal · Authors · 2023-08-08
>
> Dear Reviewer fVMQ,
>
> Thank you very much for your feedback and questions. We provide our responses below.
> ```
> Weakness 1: Marginal improvements over a very strong deterministic backbone method.
> ```
> Thanks for recognizing this. While the gain may seem marginal, we believe formulating ADA in a probabilistic manner is an important direction to pursue. The results validate its benefits, and this also provides a strong basis for further extensions accounting for other kinds of uncertainties on this probabilistic framework. In addition, we believe that a 1pp gain over strong baselines is a reasonably good improvement in domain adaptation tasks where margins are often slim.
> ```
> Weakness 2: Table/figure margins are so thin.
> ```
> Thanks for the comment. We will go through the paper and adjust the formatting to ensure tables/figures have adequate spacing and margins around them. We may have to consolidate or move some secondary tables/figures to the Appendix if needed.
> ```
> Weakness 3: Using a(b) to mean "a (or b)".
> ```
> Thanks for pointing out. We have fixed this issue.
> ```
> Weakness 4: Why in Figure 1 both green and pink go to \tau?
> ```
> Thanks for recognizing this issue. The green flow should be replaced by a purple flow, because the adversarial learning is conducted between labeled and unlabeled data. We have corrected it.
> ```
> Q1: Computational Complexity of the method.
> ```
> Thanks for the question. We have conducted experiments to evaluate the computational complexity of our method as well as other methods. Since other reviewers also raise this concern, we provide the results in the Common Response 2. Please refer to it for details.
> ```
> Q2: Why is DUC not shown in Figure 3a?
> ```
> Thank you for catching this point. We did try to reproduce the DUC results to include them in the figure, however we were unable to replicate the accuracy reported in the original paper using their provided code. Since we could not reliably reproduce their results, we decided not to include the DUC accuracy curve to avoid any incorrect presentation. Nevertheless, we are happy to provide those results for reference in the attached pdf file in the Common Response. Please refer to it if it would be helpful.
> ```
> Q3: The baseline domain adaptation performance.
> ```
> Thanks for catching this omission. We provide the baseline domain adaptation performance (i.e., DAPM without active learning) in Table R1 and Table R2 in the Common Response. It shows that without active selection and learning, our DAPM only achieves a moderate DA performance compared to modern methods that are specifically designed for UDA, given that the probabilistic backbone is mainly used for uncertainty modeling rather than accuracy. However, we are encouraged to observe that incorporating AL with our designed selection criterion significantly enhances the domain adaptation performance on all datasets. This outcome demonstrates the effectiveness of our diffusion-based uncertainty estimation, as it identifies informative samples that considerably improve overall performance.
> ```
> Q4: Why SFADA does not beat competing methods in low-data regime?
> ```
> Thanks for raising this observation. We think there are some potential reasons for this phenomenon. Firstly, in ADA, the adaptation stage has access to labeled source data. This extra supervision helps guide the model to achieve better alignments and uncertainty estimates, especially in low-data regime where target data is scarce. In contrast, for SFADA, the lack of any labeled source data removes this useful signal that facilitates adaptation when target data is scarce. Secondly, with limited data, uncertainty estimation becomes more challenging. The ADA framework is more suited to leverage our probabilistic diffusion model since it provides data with ground-truth labels to train the diffusion classifier. In the future, we will explore ways to improve SFADA performance in low-data regimes, such as using semi-supervised techniques, better leveraging unlabeled data.
> ```
> Q5: Obtain non-parametric estimates of entropy.
> ```
> Thanks for your insightful question. That is an interesting idea we have not considered. Since our diffusion model allows sampling of multiple varied predictions, we could potentially estimate entropy in a nonparametric way directly from those samples. During the rebuttal, we made a simple attempt to leverage the nonparametric entropy estimation for selection. Specifically, for each unlabeled sample, we have N predictions from the diffusion classifier, we average the entropy across the N predictions to get the final nonparametric entropy estimate for that data point, and select the ones with highest averaged entropies for annotation. The results on Office-31 can be found in Table R4.
>
> **Table R4** Results of DAPM on Office-31 using averaged entropy of N predictions for selection (ResNet-50, 5% budget).
>
> | A-\>D | A-\>W | D-\>A | D-\>W | W-\>A | W-\>D | Mean |
> |-------|-------|-------|-------|-------|-------|------|
> | 95.6|86.5|82.1| 99.2  | 82.6  | 100.0 | 92.7 |
>
> It shows that the averaged entropy on N sampled predictions does provide a good estimation of uncertainty, since the performance is better than DAPM-ENT (Table 3 in the paper), which only uses one prediction from the deterministic classifier. However, it is still below the performance of DAPM-TT. We think this only provides one approach to leveraging the samples for nonparametric entropy. We could also explore k-NN entropy, kernel density entropy or bootstrapping techniques, which also utilize the sampled predictions in a nonparametric way.
>
> Lastly, we would like to reiterate that in this work, we choose to use a t-test-based criterion since it naturally encodes the prediction variability, sample number, and cross-category ambiguity, which nicely meets the requirement for active learning. However, we agree your suggestion is worth further investigation. In the revised paper, we will add a discussion of this idea and corresponding experiments.

---

> > ### Comment · Reviewer_fVMQ · 2023-08-21
> >
> > I am happy with the responses the authors have provided. I looked through other reviews and did not see any major issues with the method to be pointed out by other reviewers.

---

### Official Review · Reviewer_3siL · 2023-07-07

**Soundness:** 3 good
**Presentation:** 3 good
**Contribution:** 3 good
**Rating:** 5
**Confidence:** 3

**Summary:**

This paper concerns Active Domain Adaption leveraging a diffusion-based adversarial probabilistic model with T-statistics for uncertainty quantification. The paper combines many existing ideas such as variational auto-encoder, diffusion model, knowledge distillation, and T-statistics. Then it proposes a new framework to improve domain adaptation performance with active learning. It shows extensive experiments with superior performances compared to various benchmark methods.

**Strengths:**

1. The paper shows a good combination of existing methods and proposes a novel framework of active domain adaptation.
2. The conducted experiments are pretty extensive and provide useful information as much as possible. It shows superior performance compared to other existing methods.

So I value the contribution of 1) proposing a new framework to combine many ideas, 2) extensive experiments.

**Weaknesses:**

The proposed framework involves many existing ideas, so combining them is somewhat complex from another perspective. I have a generic question if this combination is ideal. I also have some concerns about the sensitivity of parameters, computational costs, and active learning. Please see Questions for more details.

Minor Comments:
- Algorithm 1 in Appendix (L44-45): Eq. (??) is missing

**Questions:**

1. The paper shows a good combination of many existing methods. Because the framework also leverages the knowledge-distillation step, I wondered if the framework can be generalized to a common domain adaptation scenario WITHOUT active learning. If the framework works well without active learning, i.e., producing pseudo-labels, I think the proposed combination would be more valuable. Otherwise, we may still conclude that most of the contribution to performance improvement might come from the active learning step, even if it shows a better performance.
2. Figure 3-(b) for the parameter sensitivity shows that ADA is highly sensitive (non-convex or non-monotone) to the choice of parameters. This implies that the combined loss might conflict with each other when the source data is available. This result somewhat surprises me because more information makes the proposed framework more unstable. Could the authors reasonably explain the root of instability under a source-available scenario ADA?
3. Because it uses a diffusion classifier, I wondered how the computational cost would differ from other methods. Similar, cheaper, or expensive?
4. In active learning, diversified selection is critical. I was wondering how p-value with T-statistics acquisition can mitigate the redundant selections. For example, I imagine produced T-statistics would be very similar if two images are similar (or the same), even if it's with a diffusion classifier. Then how can we avoid the redundant selection, or how can we get a diversified selection?

**Limitations:**

The authors provided the source code in the supplementary material, but I didn't have a chance to look at the details yet.

---

> ### Author Rebuttal · Authors · 2023-08-08
>
> Dear Reviewer 3siL:
>
> Thank you very much for your detailed comments and insightful questions. We provide point-to-point responses and answers below.
> ```
> Q1: Generalization to common domain adaptation scenarios.
> ```
> Response: Thanks for the comment. It is natural to employ our method to common DA scenarios by simply disabling the selection stage, i.e., only performing the adaptation stage and the diffusion stage, as shown in Fig. 1. We provide the results of our method without active learning for common UDA in Table R1 and R2. The domain adaptation baselines for SFDA have been reported in Table 1 and 2 in the manuscript.
>
> In this case, we just aim to inference target labels in a variational Bayesian way by maximizing the ELBOs of all data points, resulting in a variational DA framework. Although the results are somewhat moderate compared to modern methods that are specifically designed for UDA, it shows that our method brings obvious improvements in the common UDA setting when compared with the baseline CDAN [22] (since we adopt conditional adversarial learning as the main adaptation technique), showing the availability of our method for common DA. In addition, it is worth noting that our framework provides flexibility for implementation. The domain adaptation baseline performance largely depends on the specific implementation for the training objectives. In this work, we leverage the knowledge-distillation to implement the second term in Eq. (12) and adopt conditional adversarial learning as the main adaptation technique, which can be improved by more elaborated pseudo labeling and adaptation techniques, as suggested by Reviewer Grmw (weakness 1). Please refer to Common Response 3 if it would be helpful.
>
> In addition, we would like to clarify that in ADA, we study how to select informative unlabeled samples for annotation and harness these labeled samples to maximally boost the domain adaptation performance. Pursuing high DA performance without active learning is a little beyond the scope of ADA research. Our probabilistic framework is primarily designed to model uncertainty and is not tasked at accuracy as deterministic methods. The large performance gains from the active learning step exactly illustrate the strength of our designed selection criterion in domain adaptation scenarios.
> ```
> Q2: Why the model is more unstable under a source-available scenario ADA.
> ```
> Response: Thanks for recognizing this point. We analyze some potential reasons for the non-convex or non-monotone surface of the model performance under the ADA setting as follows.
>
> Firstly, the source-available scenario involves more loss terms in the training objective, the additional source domain data and distribution alignment objectives like L_adv may make the optimization landscape more complex and susceptible to suboptimal solutions based on weighting hyperparameters. In contrast, in SFADA, the model solely relies on the target data and regularization losses like L_KL for alignment, reducing dependence on precise weighting.
>
> Secondly, the source data distribution itself introduces additional variability that must be accounted for, causing instability if hyperparameters are not set appropriately. In contrast, SFADA focuses more narrowly on adapting to target, with less emphasis on retaining source discriminability.
>
> Nevertheless, we think the instability does not invalidate ADA, but highlights the importance of proper hyperparameter tuning.
> ```
> Q3: Computational complexity of the proposed method.
> ```
> Response: Thanks for the comment. We have re-run our method to evaluate its computational complexity. Since other reviewers have also raised concerns about computational complexity, please refer to the Common Response 2 for more details.
> ```
> Q4: How to mitigate the redundant selections with the t-test-based criterion.
> ```
> Response: Thanks for the comment. While our current approach mainly focuses on uncertainty estimation, we would like to emphasize that it has properties that help mitigate redundant selections to some extent. Firstly, the diffusion process introduces stochasticity by generating varied predictions for the same input. This creates more diversity in the uncertainty estimates across similar instances. In other words, duplicate inputs will not necessarily have identical uncertainty. Secondly, the t-test criterion accounts for both variability across predictions and similarity of top-2 classes. Highly variable samples with closer competing classes will be favored, which is naturally distributed around the classification boundary of every class, thus enabling diversity. In visualization experiments (Appendix D.4 and D.6), we observe that our approach naturally selects a diverse range of sample classes, even though we did not explicitly impose a diversity constraint.
>
> Thanks again for raising this point. We will look to further improve diversity in the future by incorporating explicit similarity-based metrics. But we believe the current approach already promotes diversity to a certain degree by design. We will discuss this in the limitations of the work in the revised Appendix.
> ```
> Minor comments about the missing equation number.
> ```
> Response: Thanks for pointing out. The missing numbers in Algorithm 1 are 15 and 16. We have checked the Appendix carefully and supplemented all missing numbers in the revised version.

---

> > ### Comment · Reviewer_3siL · 2023-08-12
> >
> > I appreciate the authors taking the time to answer all questions during the rebuttal. I generally agree with the answers from the authors, but couldn't see further factors to increase my rating. So I keep my score as-is.

---

### Author Rebuttal · Authors · 2023-08-08

**Common Response:**

We sincerely appreciate the time and effort the reviewers have dedicated to reviewing our paper and providing thoughtful and constructive feedback. Here we provide responses to some common concerns raised by multiple reviewers. The attached pdf file contains the revised Figure 3(a) which might be useful for Reviewer fVMQ, and Table R6 and R7 which are prepared for Reviewer 1fQe.
```
Common Concern 1: The domain adaptation baseline of DAPM (without active learning).
```
Reviewer 3siL and fVMQ have concerns about the baseline performance of our DAPM in common UDA scenarios (without active learning). We provide the corresponding results in Table R1 and R2.

**Table R1** Baseline domain adaptation results on Office-Home without active learning (ResNet-50).

| Task | Ar-\>Cl | Ar-Pr | Ar-\>Rw | Cl-\>Ar | Cl-\>Pr | Cl-\>Rw | Pr-\>Ar | Pr-\>Cl | Pr-\>Rw | Rw-\>Ar | Rw-\>Cl | Rw-\>Pr | Avg. |
|-|-|-|-|-|-|-|-|-|-|-|-|-|-|
|CDAN|50.7|70.6|76.0|57.6|70.0|70.0|57.4|50.9|77.3|70.9|56.7|81.6|65.8|
|DAPM|51.9|73.1|78.4|59.6|76.3|74.9|61.2|53.5|80.2|70.2|58.3|81.3|68.2|

**Table R2** Baseline domain adaptation results on VisDA and Office-31 without active learning (ResNet-50).

| Dataset | VisDA | Office-31 |       |       |       |       |       |      |
|-|-|-|-|-|-|-|-|-|
| Task    | S-\>R | A-\> D    | A-\>W | D-\>A | D-\>W | W-\>A | W-\>D | Avg. |
| CDAN    | 70.0  | 92.9      | 94.1  | 71.0  | 98.6  | 69.3  | 100.0 | 87.7 |
| DAPM    | 80.8  | 94.3      | 93.5  | 72.1  | 97.6  | 72.3  | 99.3  | 88.2 |
```
Common Concern 2: Model size and computational complexity.
```
We realize that computational complexity is a fair concern in the study of diffusion models. To evaluate the computational overload brought by the diffusion model, we conduct a comparison among DAPM (w/ deterministic classifier), DAPM (w/ diffusion classifier), and the baseline model ResNet-50 regarding the model size and inference speed (i.e., number of samples can be handled per second during inference), on a single Nvidia GeForce RTX 3090 GPU. The results can be found in Table R3. The number of generated samples N for each instance is set to 100 as it is in the main paper. Since inputs of different datasets have the same size and preprocessing process, we do not distinguish between different datasets.

**Table R3** Model size and computational complexity of DAPM compared with baseline methods.

| Model                              | Parameters  | Inference speed |
|------------------------------------|-------------|-----------------|
| ResNet-50                          | 24.04 M     | 689             |
| DAPM (w/ deterministic classifier) | 27.20 M     | 546             |
| DAPM (w/ diffusion classifier)     | 28.68 M     | 103             |

It reveals that the inference speed of DAPM lags behind that of ResNet-50, primarily due to the additional computational overhead introduced by the calculation and sampling processes within the VAE. In comparison to DAPM equipped with a deterministic classifier, the integration of the diffusion classifier leads to a nearly 5x reduction in inference speed. This outcome is foreseeable since the diffusion model generates multiple predictions through a Markov chain with a long-time span. However, we think the compromise in inference speed is justifiable, given the enhanced capability of the diffusion classifier to capture prediction uncertainty effectively. Selecting samples based on the diffusion classifier (rather than the deterministic classifier) leads to more substantial enhancements in overall performance, as shown in Table 3. We will also include this evaluation and corresponding discussion in the revised Appendix.
```
Common Concern 3: Improved performance by incorporating other SSDA techniques.
```
Although the improvements are marginal over other strong methods at the current stage, it is worth noting that our formulation provides certain flexibility for the implementation of the training objectives, which showcases the modular nature of the framework. For instance, we use a teacher-student framework to implement the second term in Eq. (12) and adopt conditional adversarial learning as the main adaptation technique, which may limit the baseline DA performance and then affect the performance after active learning. As suggested by Reviewer Grmw, the DA component and AL component can be decoupled. We have tried to incorporate a SSDA method MME [Ref1] to implement the adaptation stage, which is also practiced by previous ADA works, e.g., [Ref2, Ref3]. Specifically, the adversarial learning loss (i.e., L_adv) is replaced by the minimax entropy loss, and other losses remains unchanged. The preliminary results on Office-Home dataset can be found in Table R4. We are still doing our best to explore the possibility of improving performance and evaluate performance on other datasets.

**Table R4** Accuracy (%) of DAPM-TT (combined with semi-supervised domain adaptation techniques) on Office-Home with 5% annotation budget (ResNet-50).

| Task              | Ar-\>Cl | Ar-Pr | Ar-\>Rw | Cl-\>Ar | Cl-\>Pr | Cl-\>Rw | Pr-\>Ar | Pr-\>Cl | Pr-\>Rw | Rw-\>Ar | Rw-\>Cl | Rw-\>Pr | Avg. |
|-------------------|---------|-------|---------|---------|---------|---------|---------|---------|---------|---------|---------|---------|------|
| DAPM-TT (w/ CDAN) | 64.2    | 85.4  | 85.7    | 69.2    | 84.2    | 83.5    | 69.1    | 63.4    | 86.0    | 77.2    | 68.4    | 88.6    | 77.1 |
| DAPM-TT (w/ MME)  | 65.1    | 85.5  | 84.9    | 72.5    | 84.6    | 83.7    | 72.4    | 64.7    | 86.3    | 77.9    | 70.3    | 88.8    | 78.1 |

[Ref1] Semi-supervised Domain Adaptation via Minimax Entropy. CVPR 2019.

[Ref2] Active Domain Adaptation via Clustering Uncertainty-weighted Embeddings. ICCV 2021.

[Ref3] Transferable query selection for active domain adaptation. CVPR 2021.

We thank you all again for your helpful guidance in revising this work.

---

### Decision · Program_Chairs · 2023-09-21

**Decision:**

Accept (poster)

**Comment:**

All reviewers recommended acceptance of this work (split between borderline, weak and strong accept). Reviewers generally found the work to be well written and technically sound. They appreciated the new method for modeling uncertainty and it's strong performance when used to help active domain adaptation solutions.